# Generating Scenarios with Diverse Pedestrian Behaviors for Autonomous Vehicle Testing

**Maria Priisalu**[1], **Aleksis Pirinen**[2,*], **Ciprian Paduraru**[3,4], **and Cristian Sminchisescu**[1,5]

[1]Lund University, [2]RISE Research Institutes of Sweden, [3]University of Bucharest, [4]Institute of Mathematics of the Romanian Academy, [5]Google Research

**Abstract:** There exist several datasets for developing self-driving car methodologies. Manually collected datasets impose inherent limitations on the variability of test cases and it is particularly difficult to acquire challenging scenarios, e.g. ones involving collisions with pedestrians. A way to alleviate this is to consider automatic generation of safety-critical scenarios for autonomous vehicle (AV) testing. Existing approaches for scenario generation use heuristic pedestrian behavior models. We instead propose a framework that can use state-of-the-art pedestrian motion models, which is achieved by reformulating the problem as learning where to place pedestrians such that the induced scenarios are collision prone for a given AV. Our pedestrian initial location model can be used in conjunction with any goal driven pedestrian model which makes it possible to challenge an AV with a wide range of pedestrian behaviors – this ensures that the AV can avoid collisions with any pedestrian it encounters. We show that it is possible to learn a collision seeking scenario generation model when both the pedestrian and AV are collision avoiding. The initial location model is conditioned on scene semantics and occlusions to ensure semantic and visual plausibility, which increases the realism of generated scenarios. Our model can be used to test any AV model given sufficient constraints.

**Keywords:** Autonomous Vehicles, AV Testing, Reinforcement Learning

## 1  Introduction

Research on autonomous vehicle (AV) models has gained momentum in recent years [1]. There exist both end-to-end AV models which make decisions directly based on visual sensor outputs [1–6], and hierarchical models which require intermediate processing (such as pedestrian detection) of sensor outputs for decision making [7, 8]. To ensure traffic safety, e.g to avoid fatal collisions [9], there is a need to evaluate the various AV models in safety-critical situations. In this paper we consider safety testing of the full pipeline of perceptive AV models – from sensor inputs (e.g. images) to steering. There exist several datasets [10–17] for developing and evaluating AV models, but manually collected data is typically gathered from traffic scenarios that seldom exhibit collision and near-collision scenarios. This shortcoming has lead to recent developments of safety-critical test case generation methods [18–32] for AV models. These existing approaches resort to simulated pedestrians which are not representative of the rich and varied behavior of real pedestrians [33] – either the pedestrian trajectories are handcrafted, or the pedestrian models are trained to behave in unnatural ways (e.g. pedestrian agents which are adversarially trained to collide with vehicles). Thus these methods may provide insufficient insights on how the AV would act in scenarios involving real pedestrians. At the same time, there exist a large number of state-of-the-art pedestrian behavior models [34–49] which learn, from real traffic scenarios, how pedestrians interact with the world.

Different from [18–32], we reformulate the problem of generating challenging scenarios as one of learning the distribution $\mu$ of pedestrian initial locations $x_0$ which are likely to induce collisions between the pedestrian and the AV, for a given pedestrian behavior model $\pi$. This reformulation allows the use of state-of-the-art goal driven pedestrian behavior models $\pi$ in AV test case generation,

---

*Work partially done while at Lund University.

5th Conference on Robot Learning (CoRL 2021), London, UK.

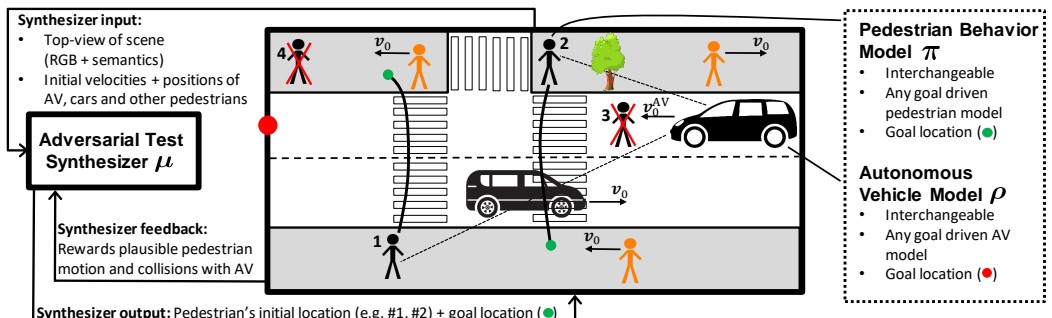

Figure 1: Overview of the proposed safety-critical test case generation model for AVs. The *Adversarial Test Synthesizer (ATS)* is trained to position a pedestrian with behavior model $\pi$ such that the induced scenario is likely to yield a collision with the AV $\rho$ (car to the right). Four example scenarios are shown. Scenarios #3 and #4 are visually implausible, as a pedestrian cannot simply appear from nowhere into the line of sight of an AV. Scenarios #1 and #2 are both plausible and challenging, as the pedestrian is close to the AV and not in the line of sight of the AV due to the occlusions.

which means that the AV can be stress tested in more realistic scenarios compared to prior works. There exist three types of pedestrian behavior models – collision seeking, collision ignorant, and collision avoiding – each of which gives rise to a distinct optimization problem in our framework. We are the first to show empirically that a non-trivial solution exists when the pedestrian model is collision avoiding. Different from previous work [18–32], we explicitly model scene semantics to learn to generate semantically and visually plausible traffic scenarios. Our model can be used to augment existing data by adding simulated safety critical pedestrians to real traffic scenarios.

In real traffic an AV can be expected to encounter pedestrians with a range of different behaviors. Some individuals follow traffic rules and plan their movements based on the surroundings; others are inattentive and take risks. Independently of the pedestrian's overall behavior, an AV should be able to avoid collisions with the pedestrian when it appears from an occluded space. To ensure this, AV test scenarios should cover the true variation of different pedestrian behaviors. Previous works [18–32] either assume that the pedestrian motion can be modelled by a simple constant velocity model, or that the pedestrian motion is adversarial to the AV. In reality however, collisions do not occur only when a pedestrian has a perfectly predictable path (e.g. constant velocity), or when the pedestrian is actively seeking to get hit by the AV (e.g. adversarial pedestrian model). Quite the contrary – most collisions occur because pedestrians are distracted, due to occlusions or noise. To alleviate the previous unrealistic assumptions on pedestrian motion in generative AV testing, we separate the problem of *finding* the pedestrian location distribution $\mu$ from the *modelling* of the pedestrian behavior. The main problem is then to find a location distribution $\mu$ such that the number of collisions between a black-box AV and a black-box pedestrian is maximal in expectation. In AV testing the proposed approach should be used with as many different pedestrian behavior models as possible, as an AV should be seeking to avoid collisions with all (even collision seeking) pedestrians.

The pedestrian location distribution $\mu$, shown in Fig. 1, is conditioned on scene semantics, distance to the AV, as well as a dynamic occupancy and occlusion map. Occlusions can cause an AV to miss a pedestrian (or vise versa) [50] and can thus cause collisions, therefore affecting the shape of $\mu$. Furthermore, $\mu$ is likely to be shaped by the scene semantics (e.g., pedestrians are more likely to reside on sidewalks than on grass) [33, 51, 52]. In previous works, test case generation for AVs has been treated as a reinforcement learning problem [20–24, 32] or as a black-box optimization problem solved by bayesian optimization (BO) [18, 21]. BO [53] cannot be used to learn $\mu$ as $\mu$ is inherently discontinuous – in realistic scenarios pedestrians can only appear from occluded spaces [51] (cf. Fig. 2). Reinforcement learning (RL) on the other hand does not assume that the policy $\mu$ is continuous, and avoids the curse of dimensionality (that occurs in classical control and planning methods) in problems, like ours, with large state spaces with unknown world dynamics [54]. We thus propose the *Adversarial Test Synthesizer (ATS)*, an RL agent which positions pedestrians in a given scene (see Fig. 1). It selects initial locations for the pedestrian according to its policy $\mu$, which is optimized to increase the number of collisions. For the ATS agent, the uncontrollable external dynamics include the scene, the AV, and all other pedestrians and cars. We model $\mu$ as a heatmap over the scene, parametrized by a deep convolutional neural network. Our pedestrian initial location model $\mu$ allows collision seeking scenario generation with any goal driven pedestrian behavior model

and any AV model. This allows for more varied and more realistic testing of the AV. We show that near-collision scenario generation with a collision avoiding pedestrian gives rise to a previously unstudied optimization problem in AV testing. We show that this problem has a solution.

## 1.1 Related Work

Previous works [21, 23, 24] have studied the generation of full pedestrian trajectories $(x_0, \ldots, x_T)$ for AV testing, such that the trajectory is adversarial to the AV $\rho$. This leads to the pedestrian only behaving in a suicidal manner. This is unnecessarily limiting as typically it is not only the AV which aims to avoid collisions. Ultimately in testing we wish to ensure that the AV can avoid collisions with adversarial as well as collision avoiding pedestrians. In [18] pedestrians are modelled with constant velocity and are initialized from a set of predefined positions. The AV is retrained in a loop with the test case generator. In [21] existing trajectories are adapted to become adversarial. The suggested method requires a varied ground truth dataset and the generated data is dependent on the variability of the existing dataset. Similarly, [25, 26, 28–31] augment existing datasets in a latent or trajectory space, which again requires a large and varied ground truth dataset.

When only testing the vehicle control of a hierarchical AV system, the set of initial locations that cause collisions can be found by the Hamiltonian-Jacobi reachability set [55]. This is not possible in our setup since we consider the full stack of the AV, not only the control problem. Moreover, in our framework the pedestrian is not necessarily adversarial to the AV, and the scene dynamics cannot be described by a differential game. Our proposed approach allows the testing of AV models with pedestrian models that are semantically aware, collision avoiding, goal reaching and articulated. We do not use robust control methods because we utilize explicit pedestrian behavior models.

There are a number of recent studies which explore visual relations in data from the AV's perspective [51, 52, 56]. Makansi et al. [51] learn a visual prior for where pedestrians and other objects can appear from the perspective of a camera mounted on an AV. A similar problem of realistic object placement in LiDAR scenes is studied in [56]. Finally, [52] show that visual cues from an on-board camera can be used to learn walkable areas in a scene. The results of [51, 52, 56] indicate that realistic data contains strong correlations between the scene's semantic structure as well as the the presence and behavior of pedestrians and stationary obstacles. We thus include such semantic cues and occlusions in our proposed pedestrian location distribution model $\mu$, as described in §2.

## 2 Methodology

In our framework, the ATS $\mu$ and the AV model $\rho$ play an indirect constrained minimax game, and no assumptions are made about the pedestrian behavior model $\pi$. Thus $\pi$ can be cooperative with either the AV $\rho$ or ATS $\mu$, or be ignorant with respect to both of these. The problem then becomes a constrained indirect three-agent minimax game with up to two agents per team. The study of the equilibrium [57, 58] is beyond the scope of this paper. However, it is clear that if the AV $\rho$, pedestrian behavior $\pi$ and pedestrian location distribution $\mu$ are unconstrained, then the minimax problem has a trivial solution. If the pedestrian is always initialized arbitrarily close to the front of the AV (when the AV's initial velocity is forward), then this will always lead to a collision. If the pedestrian and the AV always stand still or always move in opposite directions, then there are never any collisions. To avoid trivial solutions, sufficient constraints are needed.

To illustrate the minimax problem, assume that the loss functions for the AV $\rho$, the pedestrian $\pi$, and the ATS $\mu$ are respectively given by a sum of the expectation of the number of collisions and other loss components. Let the number of collisions between a given AV and pedestrian be measured by an indicator function $I$ that is 1 if a collision occurs and 0 otherwise. The AV model $\rho$ and the pedestrian location distribution model $\mu$ are learnt by minimizing the loss functions $J_\rho$ and $J_\mu$ respectively,

$$\min_{\rho} J_\rho = \min_{\rho} \left( \mathbb{E}_{\mu,\rho,\pi}[I] + f_\rho(\rho) \right) \text{ s.t. } \rho \in B_\rho \tag{1}$$

$$\min_{\mu} J_\mu = \max_{\mu} \left( \mathbb{E}_{\mu,\rho,\pi}[I] - f_\mu(\mu) \right) \text{ s.t. } \mu \in B_\mu, \tag{2}$$

where $f_\rho$ and $f_\mu$ are loss components of $J_\rho$ and $J_\mu$, respectively. And $B_\rho$ and $B_\mu$ describe the model constraints of $\rho$, and $\mu$ respectively. Equations (1) - (2) express the general optimization problem when the pedestrian behavior $\pi$ is independent of $\mathbb{E}[I]$ (for example constant velocity $\pi$). If $\pi$ is

collision avoiding then (1) - (2) together with the following equation describe the general problem

$$\min_\pi J_\pi = \min_\pi \left( \mathbb{E}_{\mu,\rho,\pi}[I] + f_\pi(\mu) \right) \text{ s.t. } \pi \in B_\pi, \tag{3}$$

where $f_\pi$ is the loss component of $J_\pi$, and $B_\pi$ describes the constraints on the model $\pi$. If the pedestrian behavior model $\pi$ is adversarial then (3) will be replaced by $\max_\pi \left( \mathbb{E}_{\mu,\rho,\pi}[I] + f_\pi(\mu) \right)$ s.t. $\pi \in B_\pi$. It is clear that the choice of the behavior policy $\pi$ changes the optimization problem and affects the solutions of $\mu$ and $\rho$. Depending on the choice of $\pi$, the set of applied constraints $B_\rho$ and $B_\mu$ may need to be adjusted to ensure that none of the models converge to a trivial solution. Previous works [18–32] have considered the cases where $\pi$ is adversarial or a constant velocity model. In our experiments we illustrate that with sufficient constraints on $\mu$, $\pi$ and $\rho$, a non-trivial solution exists for (2) when $\pi$ is collision avoiding. The classical existence conditions of a solution of zero-sum game cannot be applied [57, 59] because the problem at hand is not a zero-sum game, as the pedestrian has loss terms $f_\pi$ that are not present in the AV's loss function $J_\rho$.

### 2.1  Special Case: Three Reinforcement Learning Agents

We view the problem of learning the pedestrian initial location distribution as a reinforcement learning (RL) problem with three agents: the pedestrian, the AV and the ATS. At timestep $t \in \{0, T-1\}$ the pedestrian and the AV move in the scene by taking actions $a_t^\pi$ and $a_t^\rho$, respectively; we gather these in a joint vector $a_t = (a_t^\pi, a_t^\rho)$. The pedestrian's action is sampled from the pedestrian policy $a_t^\pi \sim \pi(.|s_t^\pi)$ conditioned on its observation $s_t^\pi$ of the scene which includes the AV. Similarly, the AV chooses actions as $a_t^\rho \sim \rho(.|s_t^\rho)$ where $s_t^\rho$ is the AV's observation of the scene which includes the pedestrian. The states $s_t^\pi$ and $s_t^\rho$ respectively contain the pedestrian's and the AV's final goal location. We join $s_t^\pi$ and $s_t^\rho$ as a vector $s_t = (s_t^\pi, s_t^\rho)$. The unknown world model $p(s_{t+1}|s_t, a_t)$ provides the transition probabilities from state $s_t$ to state $s_{t+1}$ when the pedestrian and the AV take the joint action $a_t$. The pedestrian's and the AV's actions are evaluated by the reward functions $r_\pi(s_t, a_t, s_{t+1})$ and $r_\rho(s_t, a_t, s_{t+1})$, respectively. The policies $\pi$ and $\rho$ are trained to maximize the respective expected discounted cumulative future rewards (i.e. the utility) at each state $s_t$.

We assume that the AV's initial location $y_0$, initial velocity $v_0^\rho$ and final goal location are given. Before the 0th timestep the ATS observes $s^\mu = (S, D, OP)$, where $S$ is the top view RGB and semantic images of the scene, with constant velocity predicted dynamic occupancy $D$ of the AV(calculated from $y_0$ and $v_0^\rho$), external cars and external pedestrians in the scene, and $OP$ is the elementwise product between the occlusion map from the AV's perspective $O$ and $\mu$'s prior distribution $P$. The prior $P$ is a heuristic of $\mu$ which assigns high probability to pedestrian initial locations that are close to the AV and that can lead to a collision assuming constant motion $v_0^\rho$ of the AV. The ATS agent samples an initial pedestrian location $x_0$ from the policy $OP\mu(s^\mu)$, which is the product between $OP$ and the learnable policy $\mu$. To reduce notational clutter we will in §2.1 omit the notation $OP$ from $OP\mu$ and let $\mu$ denote the policy of ATS. The pedestrian with an initial location $x_0$ is given a goal location $g^\pi$ and velocity $v_0^\pi$ such that the pedestrian's path to $g^\pi$ coincides with the AV's assuming both move with constant velocity. After sampling the pedestrian's initial location $x_0$ we simulate the pedestrian at the location $x_0$ with velocity $v_0^\pi$. Next we can simulate the pedestrian's and the AV's observation of the world $s_0^\pi, s_0^\rho$ at $t = 0$. Our aim is to find the the initial distribution $\mu$ – i.e. the policy of the ATS agent – which leads to the highest utility for the reward function $r_\mu(s_t, a_t, s_{t+1})$, where $r_\mu$ attains its highest value when the AV and pedestrian collide.

In our experiments the learnable $\rho, \pi, \mu$ are modelled by policy gradient models and share the loss

$$J = \mathbb{E}_{x_0 \sim \mu_\Theta(.|s^\mu), s^\mu \sim q, a_t^\pi \sim \pi, a_t^\rho \sim \rho, s_t \sim p(.|s_t, a_t)} \left[ \sum_{t=0}^{T-1} \gamma^t r(s_t, a_t^\pi, a_t^\rho, s_{t+1}) \right], \tag{4}$$

where $r = (r_\mu, r_\pi, r_\rho)$. The loss functions' dependence on $I$ is expressed in the different reward functions. To simplify notations let the state-action history $\tau = (a_0, s_1, ..., a_{T-1}, s_T)$, and the discounted cumulative reward $R = \sum_{t=0}^{T-1} \gamma^t r(s_t, a_t, s_{t+1})$. We can express (4) as $\mathbb{E}[R] = \int_{s^\mu} \int_{x_0} \int_\tau R(x_0, \tau) q(s^\mu) \mu(x_0|s^\mu) p_\tau(\tau|x_0) d\tau dx_0 ds^\mu$, where $p_\tau$ is the probability density function of $\tau$ given $x_0$, and $q$ is the probability density function of $s^\mu$. Let $\Omega$ be the set of allowed values for $(\tau, x_0, s^\mu)$ then for finite $T$ (4) can be rewritten to reveal the relationship between $\mu$ and $r$,

$$J = \int_\Omega q(s^\mu) \mu(x_0|s^\mu) \sum_{t=0}^{T-1} \gamma^t r(s_t, a_t, s_{t+1}) \prod_{k=0}^{t} \pi(a_k^\pi|s_k^\pi) \rho(a_k^\rho|s_k^\rho) p(s_{k+1}|s_k, a_k) d\tau dx_0 ds^\mu. \tag{5}$$

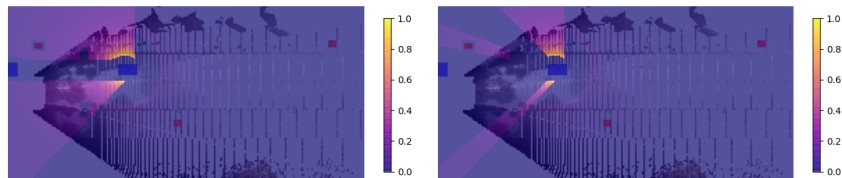

Figure 2: *Left:* A top view image of a sample prior $P$ of $\mu$. In red are other pedestrians, and in blue are cars. The prior implies a higher likelihood of pedestrian initial placement which are close to the AV. *Right:* The same prior after a multiplication with the occlusion map $O$.

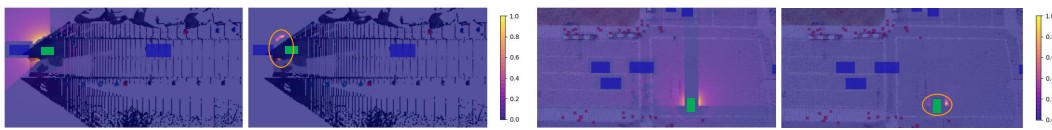

Figure 3: Top-view of two different scenarios. The AV is green, external cars and pedestrians are blue and dark red, respectively. In each of the two examples, the left image shows the prior distribution and the right image shows the final initial distribution. *Left example:* The prior $P$ induces a high likelihood for initializing a pedestrian close to the AV, but the probability map is very smeared out. The final distribution $P\mu$ is much less scattered than $P$ and more peaked close to the AV (indicated also with an external orange ellipsoid, to more clearly show where the probability mass is). *Right example:* We see a similar phenomenon as in the left example, in the dense dataset (see supplement).

From the above it is clear that $\mu$'s one step reward is $R_\mu = \sum_{t=0}^{T-1} \gamma^t r_\mu(s_t, a_t, s_{t+1})$. We use REINFORCE [60] to find $\mu$. The models $\rho$ and $\mu$ can be learnt simultaneously as shown in the supplement. If the environment model $p$ is known, we can try to find the closed form solution of (5). Using Bellman equations would then allow for a white-box treatment of the AV and the pedestrian.

## 2.2 Adversarial Test Synthesizer

The ATS is a policy gradient agent, with policy $\mu$. Its objective is to provide an initial position $x_0$ to the pedestrian agent such that the pedestrian collides with the AV. To do so the ATS needs to find locations near the AV where pedestrians and their motion are difficult to detect for the AV. To this end the ATS gets an input consisting of a top view image of the scene, the prior $P$ of $\mu$ that depends on the distance to the AV, and the temporal mapping of dynamic objects $D$. The initial distribution $\mu$ depends also on the scene semantics $S$, as ATS should learn that pedestrians are more likely to reside near certain semantic classes such as pavement. The policy $\mu$ is conditioned on the state $s^\mu$ of size $(128{\times}256{\times}C)$, where $C = 17$ is the number of channels. Due to the success of neural networks in vision tasks and the visual nature of the input $\mu$ is modelled by a two layered convolutional neural network with bi-linear interpolations and a softmax output layer (see Fig.1 in supplement). The output of the network $\mu(s^\mu)$ is a heatmap of size $(128{\times}256)$. The heatmap is multiplied by the prior $P$ (see Fig. 2) to avoid sampling $x_0$ that cannot possibly lead to a collision. To enforce visual feasibility the ATS can be required to sample $x_0$ only from locations that are occluded for the AV. This can be done by sampling $x_0 \sim OP\mu(s^\mu)$, *i.e.*the product of the occlusion map $O$, the prior $P$ and $\mu(s^\mu)$.

The ATS's reward $r_\mu$ evaluates at timestep $t$ the pedestrian's behavior at position $x_t$. Collisions with all external objects, cars and pedestrians are penalized but collisions with the AV are given a positive reward. Steps $x_t$ taken in areas often visited by pedestrians are rewarded. Steps $a_t$ towards the goal $g^\pi$ are rewarded. The reward $r_\mu$ is adapted from $r_\pi$ §2.3.

## 2.3 Pedestrian Model

The collision avoiding pedestrian behavior policy $\pi$ is the goal driven Semantic Pedestrian Locomotion model (CARLA SPL) [46]. The $\pi$ is a policy gradient agent that is trained by alternatively optimizing $\pi$ for the maximum likelihood objective of pedestrian trajectory forecasting and for the policy gradient objective of collision avoidance. The reward function $r_\pi$ (see supplement) of $\pi$ encourages motion in pedestrian dense areas with the reward term $R_{ped}$ and penalizes collisions with cars (including

the AV), other pedestrians and static objects in the reward term $R_{coll}$. The reward component $R_g$ encourages movement towards the goal location $g^\pi$, and $R_\phi$ penalizes unnaturally large motions.

The model observes a local crop $S_t(x_t)$ of size $5m \times 5m$ of the semantic labels and RGB top view image of scene $S$ and a local crop $D_t(x_t)$ of the dynamic occupancy map $D_t$. Further the state $s^\pi$ contains a history of past actions and poses taken by the pedestrian in the past $N = 12$ timesteps, the displacement to the closest car $d_t^x$ and the displacement to the goal $g^\pi$. In summary $s_t^\pi = (S_t(x_t), D_t(x_t), a_{t-1}^\pi \ldots a_{t-N}^\pi, d_t^x, \|x_t - g^\pi\|)$. The policy gradient model takes a step $a_t^\pi$ consisting of a direction and a speed. The step $a_t^\pi$ is articulated by the Human Locomotion Network.

Unless otherwise stated the pedestrian models weights are kept constant to not deviate from the learnt pedestrian motion. The CARLA SPL model is trained to avoid collisions with the external cars. The external cars have a lower average speed than the highest possible speed for $\rho$. This implies that $\pi$ expects $\rho$ to always have the same dynamics as its surrounding cars.

## 2.4 Autonomous Vehicle Model

The AV model $\rho$ is intentionally simple to illustrate the framework empirically and to avoid making constraining assumptions about the AV. The focus of this work is to show that collision avoiding pedestrian behavior models can be successfully used in autonomous AV test case generation given enough constraints on the problem. The AV is a policy gradient model with the state $s_t^\rho = (\|x_t - y_t\|, d_t, \delta_t)$ at timestep $t$; where $\|x_t - y_t\|$ is the AV's distance to the pedestrian agent, $d_t$ is the AV's distance to the closest car, and $\delta_t$ is the AV's intersection with the sidewalk. The AV's speed $c_t$ is sampled from $\mathcal{N}(\text{sigmoid}(w^T s_t^\rho + b), \sigma_\rho)$, where $w, b$ are learnt weights, and $\sigma_\rho = 0.1$. The sampled speed $c_t$ is then scaled by the maximal speed of $70km/h$. The AV's initial position $y_0$ and direction are chosen randomly among the external cars' constant velocity future trajectories.

The AV $\rho$ is assumed to have a constant direction and the policy gradient model controls the speed of the AV. Speed control can be enough to avoid collisions, as the AV can stop or accelerate to avoid a collision. Extending the AV's model to allow directional changes complicates the learning as the AV receives two conflicting objectives: to move to a goal location further ahead and to avoid collisions. The research on AVs deals with balancing such conflicting objectives, and in the future we aim to replace the minimal AV model with a state-of-the-art AV model. Replacing the current AV model with a state-of-the-art AV model requires additional constraints to avoid 0 gradients in early training (as the trained AV model may outperform the untrained ATS).

The reward function $r_\rho$ penalizes the AV for collisions with cars, people and static objects. The AV is penalized for driving on the sidewalk proportionally to the AV's overlap with the sidewalk. To motivate the AV to move, a positive reward is given at the end of the episode for the distance travelled $\|y_0 - y_T\|$. The full reward function $r_\rho$ is given in the supplementary.

## 3 Experiments

We experiment on a dataset gathered from CARLA[61]. Training data is collected from Town 1 and consists of 100 training and 50 validation scenes. The test set consists of 37 scenes from Town 2. For each scene a 3d reconstruction of RGB and semantic segmentation is created from a AV's perspective. In all experiments the scenes $51m \times 25.6m$ are voxelized into $20cm$ cube voxels. All of the tested ATS models are evaluated and trained with the *base AV model*. During initial experimentation it was noted that the AV model $\rho$ had trouble learning collision avoidance without an initializer $\mu$. The *base AV model* is trained on two scenes for 200 epochs with a $\mu$ that is trained on the training dataset for 10 epochs. A trajectory length of $T = 30$ is used to train the AV model, and $T = 100$ is used to train the pedestrian initial distribution models. Each scene is evaluated for 10 episodes with $T = 100$. During testing the pedestrian and AV models perform the mode and mean actions respectively. The action of the ATS model is sampled. The models are evaluated with three different random seeds and the average and the standard deviation (stdev) of the three runs are reported. The reported metrics are

– #. collisions - number of collisions the AV model $\rho$ has with pedestrians on average.
– $\pi$-entropy - entropy of the pedestrian policy during the length of an episode.

In Table 1 *left* the proposed pedestrian initial distribution models $OP\mu$ and $P\mu$ from §2.2 generate more than twice as many collisions (std=0.01) as sampling $x_0$ from the priors $P$ and the occlusion-

Table 1: *Left:* The proposed $OP\mu$ and $P\mu$ generate more than twice the collisions compared to the baselines; the heuristics the priors $P$ and $OP$ and the random initialization from occluded spaces *Random O*. *Right:* An ablation studying the effect of the prior during the training of $\mu$ shows that the $\mu$ is robust to changes in the prior during training as $OP\mu$ and $P\mu$ trained with the priors $OP$ and $P$ respectively, and tested with the prior $OP$, have indistinguishable collision rates (stdev 0.01).

| | Random $O$ | Prior $P$ | Prior $OP$ | $OP\mu$ | $P\mu$ | Testing prior $OP$ | $P\mu$ | $OP\mu$ |
|---|---|---|---|---|---|---|---|---|
| #. collisions | 0.06 | 0.10 | 0.09 | **0.22** | **0.24** | #. collisions | **0.21** | **0.22** |
| $\pi$-entropy | 0.85 | 0.60 | 0.65 | **0.25** | **0.24** | $\pi$-entropy | 0.29 | **0.25** |

Table 2: Collision rates of the $P\mu$ model trained with collision avoiding, distracted, collision seeking and constant velocity pedestrians. The pedestrian model does not affect the collision rate of the proposed $\mu$, as long as the pedestrian model is not the constant velocity model.

| | Collision avoiding SPL | Distracted SPL+$\epsilon$ | Adversarial SPL A. | Adversarial STPN | Constant velocity HLN |
|---|---|---|---|---|---|
| #. collisions | **0.21**($\pm$0.02) | **0.22**($\pm$0.03) | **0.22**($\pm$0.01) | **0.19**($\pm$0.01) | 0.11($\pm$0.02) |
| $\pi$-entropy | 0.29($\pm$0.01) | 0.25($\pm$0.02) | 0.029($\pm$0.001) | 0.53($\pm$0.03) | **0** |

masked prior $OP$. This confirms that $\mu$ learns and improves beyond the initial prior distribution, and that $OP\mu$ produces more collisions than the hand-designed heuristics $P$ and $OP$. The baseline *Random O* the random initialization of pedestrians from occluded spaces with 360°field of view has the lowest collision frequency. This is likely because occluded spaces may be far from the AV. The proposed $OP\mu$ has a lower $\pi$-entropy than the prior $OP$ suggesting that $OP\mu$ has learnt to initialize the pedestrian such that the pedestrian's direction of movement is as predictable as possible. With low $\pi$-entropy $\mu$ has more control over $\pi$'s trajectory. To the *left* in Fig. 3 the prior $P$ and the corresponding scene's $P\mu$ distribution are visually compared. The $P\mu$ has learnt decisively to initialize the pedestrian near the AV, and with a higher probability towards the sidewalk than the road.

In Table 1 *left* the models $OP\mu$ ( i.e. $\mu$ trained and tested with the prior $OP$) and $P\mu$ ( i.e. $\mu$ trained and tested with prior $P$) showed no significant difference. Showing that a 90°view occlusion map does not significantly affect $\mu$. Further applying the occlusion mask $O$ only in testing does not affect the number of collisions, as seen when comparing $P\mu$ to $OP\mu$ in Table 1 *right*. This suggests that curriculum learning may be used to enforce larger changes to the prior $P$ to facilitate 360°field of view occlusion masks (for LiDAR data). A visual comparison of $P$ and $OP$ can be seen in Fig. 2.

In Table 2 the following pedestrian behavior policies are used to train $\mu$,
– *Collision avoiding SPL* - the goal reaching collision avoiding pedestrian model described in §2.3
– *Distracted SPL+$\epsilon$* - a distracted SPL pedestrian. With a 0.3 probability at each timestep the pedestrian will not notice the AV for $m \sim Poisson(2)$ timesteps.
– *Adversarial SPL A.* - an adversarial SPL agent. The *SPL* model that is finetuned with the $R^\mu$ reward. *SPL A.* is trained simultaneously with $\mu$ (see supplementary Algorithm 1 with $\alpha_\rho = 0$).
– *Adversarial STPN A.* - an adversarial agent that has the Semantic Trajectory Policy Network architecture [46] i.e. the SPL architecture without the Human Locomotion Network (HLN). The *STPN A.* is trained from random weights simultaneously with $\mu$ to maximize the the number or collisions ($R_{STPN} = I$ from §2). *STPN A.* is not trained to maximize the negative log-likelihood of pedestrian trajectories like the *SPL* models, and it is the only model without locomotion.
– *Constant velocity CV* - constant velocity motion articulated by [62]. The agent moves towards the goal with a speed drawn from a Gaussian with $\mu = 1.23\text{ms}^{-1}$ and $\sigma = 0.3$ [63].

The models $P\mu$ trained with the collision avoiding *SPL*, the distracted *SPL+$\epsilon$*, the adversarial finetuned pedestrian policy *SPL A.* and the adversarial *STPN A.* (most similar to previous work) are on-par, showing that $\mu$ can learn to control the collision avoiding *SPL* as well as an adversarial pedestrian model. The collision seeking *STPN A.* does not outperform the collision avoiding *SPL* likely due to *STPN A.*'s high entropy that makes *STPN A.* hard to control for $\mu$. *STPN A.* has no motion priors and can get hit by the AV with motions that have a low likelihood in real pedestrian trajectories, such as zigzagging in the middle of the road. Even though the *CV* model is the most controllable, the initializer trained to control *CV* results in the lowest collision rate because the AV has an easy time avoiding collisions with the *CV*. The $\mu$ trained on *SPL+$\epsilon$* distracted pedestrian could be expected to have a higher collision rate than *SPL*, as $\pi$ has a noisier estimate of the AV's position. Unfortunately $\mu$ does not learn to utilize this unnaturally unstructured (and thus unpredictable) noise.

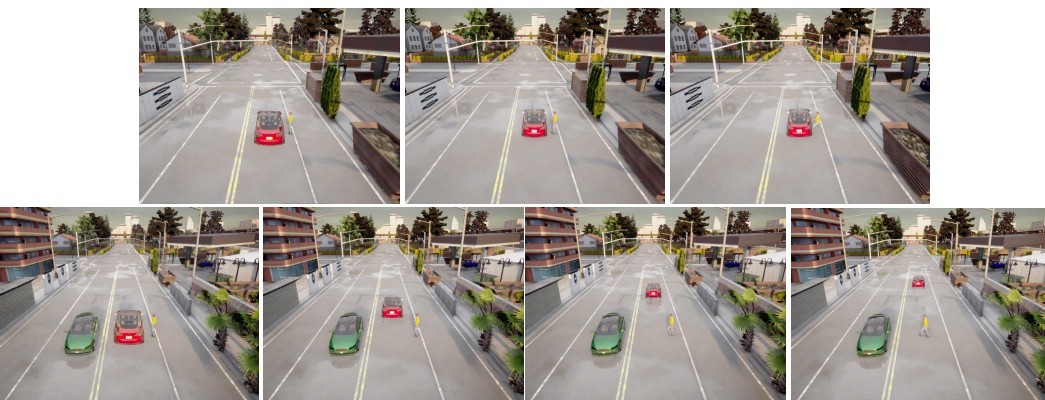

Figure 4: Sample trajectories of the *Simultaneous-$\mu$, $\rho$* model, sub-sampled at 5 frames from frame 0. *First row:* The AV changes speed thus causing the pedestrian to incorrectly estimate the AV's motion and walk into the AV. *Second row:* The pedestrian waits for the AV to pass before crossing the road.

This illustrates the need for a realistic noise model. Natural noise in the pedestrian's observation of the AV could be expected to be more structured, for example high noise levels are expected near occluded spaces. Some sample trajectories of the simultaneously trained $\mu$ and $\rho$ (see supplement) are shown in Fig. 4 - a collision prone initialization, and an initialization that does not lead to a collision because the AV speeds away.

## 4   Conclusions and Future Work

We are the first to utilize state-of-the-art pedestrian forecasting models in generative AV testing. We have proposed a general framework that is capable of stress testing the collision avoidance of AVs with a wide range of pedestrian behavior models. In practice we wish to ensure that an AV can avoid collisions with all pedestrians (intoxicated, law-obedient, children etc.), and thus should test the AV with as many different pedestrian behaviors as possible. Our empirical evaluations show that a goal driven pedestrian model with any behavior can be used in this framework. This is a significant result, as no prior work has shown that a collision avoiding pedestrian model can be used to generate collisions with a collision avoiding AV. To achieve this, we have proposed the *Adversarial Test Synthesizer (ATS)* which, given any goal driven pedestrian model, learns the pedestrian initial location distribution $\mu$ that maximizes the expected number of collisions with a given AV. The ATS is modelled by a neural network which receives as input the top view image of the scene, the scene semantics, the occupancy of dynamic objects, and outputs a distribution $\mu$ over pedestrian initial locations. We have shown that $\mu$ can learn to adversarially position a collision avoiding pedestrian model that has been trained on ground truth pedestrian data and obeys human locomotive dynamics. Our work, for the first time, shows that generative models of AV test scenarios can utilize state-of-the-art pedestrian motion models instead of the typically used models which do not resemble real pedestrian motion. Stress testing AVs with state-of-the-art pedestrian forecasting models decreases the statistical difference between tested and real pedestrian behaviors, which could reduce the likelihood of real life AV crashes.

We have shown that a learnable pedestrian initial location distribution $\mu$ exists for stress testing a basic AV model. Ultimately we wish to extend the result to state-of-the-art AV models. Since the model $\mu$ treats the AV and the pedestrian agent as black-boxes, $\mu$ can be trained to adjust to the dynamics of a more sophisticated AV as is. However, finding a non-trivial solution will require a careful readjustment of the choice of sufficient but realistic constraints. The problem can be constrained spatially by tight streets, occlusion dense scenes, lack of space due to traffic density, or by setting a limit on the pedestrian's maximal distance to the AV. Alternatively, the pedestrian model can be constrained by adjusting the noise level of the pedestrian's internal prediction of the AV's future motion, the noise level of the pedestrian's observation of the AV, or the pedestrian's goal, dynamics or personality. Similarly, the noise in the AV's observation of pedestrians and other traffic participants and their motion should be high enough to lead to collisions. As seen in the experiments, unstructured noise cannot be utilized by $\mu$, thus careful modelling of the noise of the chosen AV's observations is required. In future work the pedestrian's internal prediction of the AV's motion could be impaired with a psychologically or physiologically inspired noise process.

**Acknowledgments**

This work was supported by the European Research Council Consolidator grant SEED, CNCS-UEFISCDI PCCF-2016-0180, and the Swedish Foundation for Strategic Research (SSF) Smart Systems Program. We would also like to thank the anonymous reviewers for their useful comments.

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
