# OpenReview forum: "Generating Scenarios with Diverse Pedestrian Behaviors for Autonomous Vehicle Testing"
_robot-learning.org/CoRL/2021/Conference — CoRL2021 Poster_

### Official Review · Reviewer_s5dQ · 2021-07-17

**Originality:** Very Good
**Technical Quality:** Good
**Clarity Of Presentation:** Fair
**Impact:** 3

**Recommendation:**

Weak Accept: I recommend accepting the paper, but will not argue for my recommendation if the majority of other reviewers have a different opinion.

**Summary:**

This paper proposes a reinforcement-learning based approach for learning a distribution over initial pedestrian locations which are challenging for a nearby autonomous vehicle. The core idea is to separate learning the distribution of initial pedestrian locations that lead to collisions from the human behavior model. This approach is compared in the CARLA simulatior with alternative methods such as randomly generating pedestrian locations or generating them in occluded spaces.


**Issues:**

- Please provide further justification for the proposed idea of decoupling the initial states from the human behavior policy (and the autonomous vehicle model). See the detailed comments above.
- Please clarify the writing and notational consistency so the proposed method can be more clearly understood.

**Reviewer Expertise:**

Very good: Comprehensive knowledge of the area

**Strengths And Weaknesses:**

Strengths:
- The problem of interest in this paper -- generating challenging simulated scenarios for autonomous vehicle testing -- is critically important and underexplored
- The idea of characterizing a distribution over initial pedestrian states which would induce a challenging scenario for the AV is interesting

Weaknesses:
- The proposed approach of decoupling the unsafe initial state distribution from the human behavior model requires more justification. Furthermore, there are too many confounding variables in the design of the proposed approach which make the utility of the overall pipeline difficult to evaluate.
- It is unclear how realistic the proposed approach is for modern autonomous vehicles. More details provided below in the detailed comments section.
- The clarity of the writing (both from a technical and also narrative perspective) could be improved.

Detailed comments:

Section 1:
- Figure 1. Why does the AV input not include the history of the agent’s movement/poses (if they are not occluded)? This seems key since a core ability of the AV is re-planning and updating its predictions conditioned on the history of the human.
- Figure 1 caption states that Scenarios #3 and #4 are illegal, but that seems like a misnomer. They aren’t illegal, but rather infeasible (under the laws of physics) for a pedestrian to “simply appear from nowhere into the line of sight of an AV”.
- “We have strong reason to believe that the optimal pedestrian distribution \mu is not easy to model…” This sentiment would be strengthened with a more substantive, quantitative definition of “easy to model”. For example, does this refer to the difficulty in choosing the correct features which influence the pedestrian distribution? Does this refer to the dimensionality of the distribution?
- “... the pedestrian location distribution \mu is a learnable stochastic policy” The term policy canonically refers to a mapping from states to a (distribution over) actions. With this definition, it is unclear how a distribution over initial states is a policy. Clarifying this use of language would improve this sentence.
-  “This allows us to model \mu as a convolutional neural network \mu_\Theta” The use of a neural network to model the initial distribution is not well motivated by the preceding paragraphs. For example, alternative methods for computing initial conditions from which safety-critical scenarios can happen include robust-control methods like Hamilton-Jacobi reachability, which compute challenging initial states via the solution to a dynamic programming problem. In the stochastic realm of approaches, the distribution over initial states could be parameterized by a mixture of Gaussians instead of a neural network, perhaps even enabling more sample efficiency. Discussing why a neural network model is essential for representing \mu would strengthen the motivation and technical approach of this work.
- In the introduction, works which model adversarial (i.e., “suicidal”) pedestrians are referenced and this work seeks to differentiate itself from these works. However, in the second to last paragraph of Section 1, the pedestrian behavior “can be adversarial to the car or the pedestal location distribution \mu, or even not directly cooperative to either side.” This appears to be modelling an adversarial setting, which is precisely what the introduction was claiming was an undesirable modelling paradigm. Clarifying this is essential for understanding the difference between the proposed method’s philosophy and existing work.

Section 2:
- “When the objective J is maximized…. Then both pi and mu become adversarial to the car’s objective J_\rho”. Notationally this sentence could be made more consistent: the car’s objective is referred to as J^p (not J_\rho) throughout the rest of the text. The reason behind why this statement is true is unclear due to the fact that J is not described, the relation between J and J_\rho is not established, and it is unclear where the adversarial nature of pi and mu come into play.
- “This leads to the pedestrian only behaving suicidal.” Grammatical reworking of this sentence: “This leads to the pedestrian behaving in a suicidal manner.”
- “To allow for both, we separate nu explicitly into the initial pedestrian locations … and the pedestrian behavior pi.” This is one of the core ideas behind this paper. However, it is not clear why this decoupling should work. More specifically, the same initial pedestrian locations could (a) cause collisions with the AV or (b) not cause collisions with the AV under different pedestrian behavior models and different autonomous vehicle planning/control architectures. For example, a human standing in front of a car is doomed for collision if the human moves towards the car and the car continues forward. But, a human standing in front of a car isn’t doomed for collision if the car reverses and the human runs away. This implies that the distribution over “dangerous” initial pedestrian locations is conditioned on the human policy as well as the autonomous vehicle policy, i.e., mu(S, O, D, P, pi_human(s, g), pi_AV(s)) where pi_human is the human’s policy and pi_AV is the autonomous vehicle’s planner/policy. The justification behind why decoupling the pedestrian behavior and the dangerous initial locations is a reasonable modelling assumption should be improved to increase the overall quality / clarity of the technical contribution.

Section 3:
- “...pi is constant or independent of E[I].” This sentence is missing a period at the end.
- “If the pedestrian behavior model \pi … then the third equation will maximize…” There are only two equations stated in Section 3, so it is unclear what third equation this sentence is referring to.
- p(s^\rho_t) is the goal-driven car model, but it does not appear to be goal-dependent like the pedestrian model. Since an initial pedestrian state’s safety depends not only on where the pedestrian is trying to move but also where the AV is trying to move, it appears critical that the car’s model also captures how the AV behaves differently as a function of it’s own goals. Furthermore, it appears that the car model only depends on the state of the ego vehicle, s^p_t. Thus, it is not possible for the autonomous vehicle to react to / predict the pedestrian since it never has access to the pedestrian’s state, which is an unrealistic modelling assumption of an AV autonomy stack.
- The external world dynamics are denoted by p(s_t+1 | s_t) but this appears to be an autonomous system without any input. Since the text states that these dynamics also include the AV and the pedestrian and any other traffic participants, then it must also include the actions of all other agents too: i.e., p(s_t+1 | s_t, a_pedestrian, a_AV, a_other_traffic_participants).
- There is a notational inconsistency in equation (3) and (4) when referring to a^p_t and a^c_t -- I believe these should be a^pi and a^\rho respectively.
- In equation (4) is the policy pi(a^p_t | s_t) the same pedestrian behavior model pi(s^pi, g^pi)? The notational inconsistency makes it unclear to understand what this part of the equation represents. A similar question can be asked of the quantity p(a^c_t | s_t).

Section 3.3:
- “The car model is intentionally simple… The car is a policy gradient model…” Since the safety of a pedestrian’s initial condition and the car’s policy are inextricably intertwined, the safety assessment of the initial conditions only holds under an appropriate choice of the car model. Realistic autonomous vehicle stacks are incredibly complicated. To assess the impact / utility of the proposed framework, a more realistic car model should be discussed or, ideally, evaluated.

Section 3.4:
- “It should be noted that the CARLA … is trained to avoid collisions on a dataset where cars have much lower average speed…. This provides sufficient constraints on the collision avoidance abilities of \pi..” This claim is not well justified. Since there is a model mismatch between the simulated human’s policy with respect to other cars, then the human policy may not appear as collision-avoiding as claimed. For example, if the human policy only had to avoid slow-moving cars, then the human policy may not run out of the way of an on-coming faster vehicle, the way a truly cooperative collision-avoiding agent would. This is a confounding variable in the setup of the simulator, and should be either removed as a confound (by using an appropriate pedestrian policy that can truly react to the car in a cooperative way) or it’s effects in the construction of distribution over unsafe initial states should be further discussed / controlled for.

Section 4:
- There is a typo in Table 1 -- OP should be PO.
- The text references a distribution P\mu-D but this is not described in Table 1. Perhaps this is a typo on the far-right distribution P-D?
- “... m ~ Po(2)” does Po refer to probability distribution? And what does the argument of 2 mean? Please clarify.
Section 5:
- Typo “mdoel” should be “model” and “coo-operative” should be “cooperative”.


**Summary Of Recommendation:**

The problem of interest in this paper is very important. However, as currently written, the proposed approach and results are too preliminary and difficult to assess. This paper would be better contextualized as a Blue Sky paper focused on (1) clearly outlining what it takes to do “good” simulation / stress testing of safety-critical scenarios and (2) reworking the proposed approach as a first step towards the simulated, automated stress testing of AV safety systems. Clearly describing (1) is something that is still lacking in the literature, and could be a contribution on its own; after this, concrete instantiations of AV safety testing frameworks -- like perhaps the one proposed -- can be better contextualized.

================
After the discussion period, I have decided to increase my score to "weak accept".

---

> ### Author Response · Authors · 2021-08-31
> **Reply to the various comments**
>
> We have thoroughly edited the paper.
>
> The Bellman equations (i.e. the recursive reformulation of (5)) can be solved leading to a white box treatment if all of the world dynamics $p$ are known, but this is not the case. The suggested Hamilton-Jacobi reachability would involve solving the Hamilton-Jacobi-Isaac equation which is the continuous time equivalent to Bellman equations. Secondly the Hamilton-Jacobi-Isaac equations assume a zero sum game, which would imply that the AV cannot minimize a cost function with behaviour measures that not related to pedestrian's (for example avoiding collisions with other entities). Using a robust control method assumes that we know the world dynamics. The world dynamics is unknown as the pedestrian model (and possibly the AV in models such as Learning by Cheating) act on high dimensional visual input. The dynamic model of the external pedestrians and cars is assumed to be unknown to allow the method to be applicable for data augmentation. Finally the pedestrian and cars may be stochastic (for example to model a pedestrian who suddenly changes direction).
>
> The pedestrian initial distribution model $\mu$ observes a top view image of the scene, to allow the model to learn the scene geometry and semantics without enforcing hand-crafted rules. It is standard to utilize neural networks to model visual input, as neural networks have been shown to outperform all hand designed features in object detection and semantic segmentation in well establised benchmarks. The network structure of $\mu$ is inspired by the SPL-model. In initial architecture search we experimented with Gaussian mixture models, but noted that the model failed to improve over the prior.
>
> We would like to clarify that $\mu$ is never considered to be independent of rho and $\pi$. Please note that the equations (1,2,3) show the dependence of the three models on one another.  However note that $\mu$ does not need to observe the policy of $\pi$ and $\rho$, but learns to estimate the relevant dynamics. Think of this as a driver being able to predict a pedestrian’s future trajectory from previous experience, and not by knowing this individual’s internal plan. Avoiding observing $\pi$ and $\rho$ explicitly means that no assumptions are made on the action space of the AV and the pedestrian, making $\mu$ easy to utilize with any pedestrian and AV model. Secondly please note that $\mu$ observes the AV’s initial velocity and goal in dynamic occupancy map $D$, and the prior $P$. The pedestrian’s initial velocity is selected to point towards the goal that crosses the AV’s future trajectory.
>
> The AV’s state is assumed to include an observation of the pedestrian. We attempt to make no further assumptions about $s^\rho$ because the AV may observe the pedestrian through a LiDAR scan, an image, etc.
>
> We tested a number of AV architectures observing the pedestrian’s velocity as a vector and as an angular input, but this decreased the AV’s performance. We did not wish to increase the complexity of the AV as this would simply mean repeating the efforts of existing AV models. Instead we show that it is possible to find a collision prone pedestrian distribution for collision avoiding pedestrian models. We tried to finetune the AV model further with the additional input of the pedestrian’s velocity, but this did not significantly change the collision rate (0.22).
>
> We have constantly kept modern AV methods in mind when designing the model, and there is no reason why $\mu$ would not be applicable for a state of the art AV model. However note that if the problem has insufficient constraints then a trivial solution may be found, namely that the AV and the pedestrian never collide. To ensure collisions observation noise must be carefully modeled both for the pedestrian and the AV. In the section Conclusion and Future Work we discuss the possible adjustments to the behaviour constraints $B_\rho$ and $B_\mu$ that may be needed when the AV is better at avoiding collisions.
>
> The pedestrian model is trained to avoid collisions with the external cars. This is not an unrealistic scenario, as the pedestrian expects the AV to move like the other cars.  The collision avoiding policy is never claimed to be perfect, and it in fact does not need to be. The AV needs to be tested also on pedestrians that misjudge the AV’s motion but still attempt to avoid it. If both the AV and the pedestrian are perfect at avoiding collisions, then no collisions occur. We are looking for the solution of a constrained problem where one of the constraints may be the pedestrian not being perfect at avoiding collisions. We illustrate in this paper a sample set of constraints on the problem that provide a non-trivial solution. Further finetuning the pedestrian on a AV speed augmented dataset (with speeds from 0 to 70km/h), and then training a $OP\mu$ model for the pedestrian resulted in an insignificant change in the number of collisions (0.22).

---

> > ### Comment · Reviewer_s5dQ · 2021-09-03
> > **Really fascinating problem, still have some continued concerns**
> >
> > I want to thank the authors for their time and effort in replying to my comments and elaborating on their approach. I still believe that the problem studied in this work is very timely and truly interesting. I also believe that the approach is an interesting learning-based perspective on safety verification for autonomous vehicles. However, I want to push back a bit on some of the high-level rebuttal comments mentioned in the AC comment, as they highlight the sources of my discomfort in confidently accepting this work. At the highest level, I am still somewhat unconvinced that this problem is best solved via the proposed ML approach.

---

> > > ### Comment · Reviewer_s5dQ · 2021-09-03
> > > **(continued)**
> > >
> > > First, I do not completely agree with the statement that “A learning approach is needed to avoid assuming that the AV’s, pedestrian’s and the external traffic entities' dynamics are known.” Existing AV automation stacks do use high-quality approximate AV dynamics models based on first-principles, and similar models exist for pedestrian dynamics as well that do not necessarily require state spaces to be defined in terms of image pixels. In fact, I would argue that the core challenge in determining the set of initial states from which, if the pedestrian starts, it is guaranteed to collide with the AV comes from not knowing the appropriate human behavioral models at validation/stress-testing time. Perhaps these behavior models can be data-driven (as they often are), but the need for an ML-based verification method could be significantly strengthened in the paper. For example, if we have access to to a (possibly learned) human policy and the AV’s policy (both as black-box models), then for a given scene, one could naively iterate through all (human, AV) pairs of initial conditions, forward simulate where each agent ends up by querying the black-box models, and then recording which (human, AV) initial states led to collision. Or, if you are given a pedestrian behavior model and assume that the AV either (a) follows some (possibly black-box) planning model or (b) the AV can take any dynamically-feasible controls, then you can also quantify the set of unsafe initial (joint) positions of the AV and pedestrian via dynamic programming and robust-control methods (as mentioned in my original review) by restricting the set of feasible controls for the human pedestrian according to their behavior model. Given the simplifying assumptions of the human and AV models used to evaluate this work, either of the two approaches from above could be possibly implemented and compared to as valid alternatives. This is related to Reviewer 7Zap’s comment on “If the goal is to generate test cases, the result may need to be compared with manually generated test cases either heuristically or by filtering randomly generated scenarios. If the goal is to generate collision-prone smart agents then how it is better than a model predictive system that is designed to collide with another agent” is a valid one that still has not been completely addressed for me. For this reason, I am still somewhat unconvinced that the proposed RL approach to this problem is completely necessary and/or well-justified.
> > >
> > > Related to the comment above, since the core problem of this work is to “find a pedestrian location distribution µ such that the number of collisions between a black-box AV and a black-box pedestrian is maximal in expectation,” I believe it is important to augment the related work with methods from the non-learning-based verification community (e.g., the zero-sum differential game formulations of the pedestrian and AV) since these tools are well-established methods for addressing (the worst-case formulation of) the proposed problem. While I wholeheartedly agree with some of the author’s points about the drawbacks of these formulations (e.g., not scaling well to high-dimensional inputs like images), it is nevertheless important to describe that the problem at hand is not a new one, discussing pros/cons of different community’s approaches, and the applicability (or lack thereof) of non-learning-based methodologies to the domain studied in the proposed work. This nuanced treatment of how different communities have tackled this problem would not only flesh-out the context of this work, but also allow the authors to clearly and precisely state why their learning-based approach is necessary to solve this problem for modern AV’s.
> > >
> > > Finally, I do not completely agree with the statement that  “considering the problem of placing out a single pedestrian to test the AV is a harder problem than placing out multiple pedestrians out of whom at least one should cause a collision for the AV.” While yes “multiple pedestrians constrain each other and the AV’s future motion more than a single pedestrian”, the interaction between multiple pedestrians over time (a) induces trajectories (and occupied states) that can make collision-avoidance harder for the AV and (b) in the worst-case analysis, multiple pedestrians can in fact “collaborate” in an adversarial fashion to cause collision with the AV (e.g., consider the scenario where the pedestrians are malicious teenagers who decide to run towards and block/trap the AV). Mentioning this as future work would be valuable to put into the paper.
> > >
> > > For the above reasons, I do not feel comfortable changing my recommendation at this time.

---

> > > > ### Author Response · Authors · 2021-09-05
> > > > **On the differences of the control and the RL problem**
> > > >
> > > > In our reply to the AC in “A learning approach is needed to avoid assuming that the AV’s, pedestrian’s and the external traffic entities' dynamics are known”, we meant that we may know the pedestrian agent’s and the AV’s motion models, but we cannot describe the dynamics of the full traffic scenario without knowledge of the traffic process which we attain as an (possibly LiDAR) image. Specifically for a general traffic scene (when for example augmenting existing data) we cannot forecast the number of pedestrians in the scene (new pedestrians can enter the scene after the first timestep), the behaviour of each individual pedestrian, the number of cars, their location and their movement direction, and unexpected obstructions (animal or a child running on the road, an object that has fallen onto the road from a car, a collision among the external entities, change in traffic lights, etc).
> > > >
> > > > Our framework allows the testing of any realistic AV in the vicinity of any realistic goal driven pedestrian behaviour model. The s5dQ’s suggested planning approach would not allow the testing of AV[C-F] and pedestrian behaviour [31,33,38,39,40,41,42] models with visual inputs.
> > > >
> > > > A planning problem is solvable as a Markov Decision Process (MDP) when the probability distribution of the state transitions ($p(s_{t+1}|s_{t},a_t)$) is known. The reviewer suggests that $p(s_{t+1}|s_{t},a_t)$ could be approximated by some dynamic models of pedestrians and AVs. It is well known [A,B] that solving the problem (2) with RL the Adversarial Test Synthesizer (ATS) learns from the given data and simulations to solve the MDP when $p(s_{t+1}|s_{t},a_t)$ is unknown, where the state description $s_t$ may contain images. RL is the suggested solution by Bertsekas in [A] if the state space is large and the environment dynamics are unknown. Videos (possibly 3D) are the only complete description of the environment and thus it is preferable to use the available visual data of the scene and not approximative dynamic models for the external pedestrians and cars. It is disadvantageous to use approximative models for the external entities in planning their motion (for T timesteps), since the true motion is available from simulations and the dataset. Using approximative models in planning can be expected to lead to cumulative errors.
> > > >
> > > > The suggested naive approach (i.e. exhaustive search), would require us at least 5345h=223 days to evaluate the training set only, as each scene consists of 256\times128 pixels, there are 100 scenes and each should be evaluated for 100 timesteps at a framerate of 17 frames per second. Thus $256\times128\times100\times100\times 1/17 \approx 19 275 294$s. If the pedestrian policy is sampled (as is done in training) this would require at least 10 samples per pixel location thus requiring around 53453h. Clearly a method that generalizes across scenes (such as ours) is desirable instead. We are happy to compare our method to a publication that presents the full models used, and is scalable to multiple scenes, and re-creatable in a reasonable amount of time in our experimental suite. Note that [A] suggests that an RL approach should be used instead of exhaustive search to avoid the curse of dimensionality.
> > > >
> > > > Dynamic programming has complexity O(n^2), with $n=256\times128\times100\times100\approx3.78\times10^8$ states for the training set, where $n^2\approx10^{17}$. In [A] an RL approach is suggested as the preferred approach to dynamic programming in large state spaces to avoid the curse of dimensionality.
> > > >
> > > > Using the robust control methods requires a two agent zero-sum game which we do not have. The pedestrian agent cannot be modeled as the uncertainty in the control of the AV, as real pedestrians themselves are agents with their own goals and reward functions. We are not studying the worst case scenario for the AV with a fictive agent that does not resemble a true pedestrians (i.e. a pedestrian that is always taking opposing actions to the AV like in [G, 14, 16, 17]), but the scenarios that lead a given realistic collision-avoiding pedestrian agent to collide with the AV.
> > > >
> > > > In our problem the traffic situation is the state of the system. We cannot express the evolution of the state of the traffic deterministically and continuously. Therefore we cannot express the evolution of traffic situations as a system of differential equations. Thus we cannot use the differential game approach and also we cannot use the Hamilton-Jacobi reachability.
> > > >
> > > > Please note that we do compare our model to the prior $P$ that is a heuristic that defines the points $h$ (see supplementary for details) from which the pedestrian agent and the AV would collide if both always had a constant velocity (not this is an incorrect assumption for both) and there were no external changes in the environment.

---

> > > > > ### Author Response · Authors · 2021-09-05
> > > > > **continued**
> > > > >
> > > > > We are happy to discuss any particular publications you have in mind in the related work. The problem of generating collision scenarios when both the pedestrian and the AV are collision avoiding is the novel problem. Please note that there is no non-learning method of expressing the pedestrian initial location distribution as a function of the scene geometry and semantics in a manner that does not depend on hadcrafted rules. By noting the previous work in test case generation for an AV with learning methods was an attempt at showing that even specifically the learning problem is well studied, but we are happy to extend the related work with non-learning methods.
> > > > >
> > > > > Finally regarding the placement of multiple pedestrians into the scene such that the positions lead to collisions, both of your illustrative examples (a) and (b) show cases where the AV is more prone to collide in the presence of multiple pedestrians than in scenarios with a single pedestrian. These examples illustrate cases where multiple pedestrians have more initial locations that lead to collisions than the one-pedestrian problem. It is clear an initial location $x_0$ in the one-pedestrian agent set up that leads to a collision is also a solution for the multi-pedestrian agent set up assuming that one pedestrian is initialized at $x_0$ and the other pedestrians are initialized at a large enough distance to $x_0$ and the AV. Therefore having shown that an initial location distribution exists for one controllable collision avoiding pedestrian we know that an initial location distribution exists for multiple controllable initial location distribution pedestrians. The hardness of the problem only refers to the number of solutions, and not to how easy the solutions are to find in practice. Note that a multi-agent RL game can be solved by treating each individual player of the game separately[G].
> > > > >
> > > > > [A] Bertsekas, Dimitri. Reinforcement Learning and Optimal Control. Athena Scientific, 2019.
> > > > >
> > > > > [B] Sutton, Richard S., and Andrew G. Barto. Reinforcement learning: An introduction. MIT press, 2018.
> > > > >
> > > > > [C] Abbas Sadat, Mengye Ren, Andrei Pokrovsky, Yen-ChenLin, Ersin Yumer, and Raquel Urtasun. Jointly learnable behavior and trajectory planning for self-driving vehicles. In IROS, 2019. 5, 6, 7, 8, 15
> > > > >
> > > > > [D] Wenyuan Zeng, Shenlong Wang, Renjie Liao, Yun Chen, BinYang, and Raquel Urtasun. Dsdnet: Deep structured self-driving network.CoRR, abs/2008.06041, 2020. 2, 7
> > > > >
> > > > > [E] Chen, Dian, et al. "Learning by cheating." Conference on Robot Learning. PMLR, 2020.
> > > > >
> > > > > [F] Toromanoff, Marin, Emilie Wirbel, and Fabien Moutarde. "End-to-end model-free reinforcement learning for urban driving using implicit affordances." Proceedings of the IEEE/CVF Conference on Computer Vision and Pattern Recognition. 2020.
> > > > >
> > > > > [G] Zhang, Kaiqing, Zhuoran Yang, and Tamer Başar. "Multi-agent reinforcement learning: A selective overview of theories and algorithms." Handbook of Reinforcement Learning and Control (2021): 321-384.

---

### Official Review · Reviewer_UAuu · 2021-07-23

**Originality:** Very Good
**Technical Quality:** Good
**Clarity Of Presentation:** Fair
**Impact:** 4

**Recommendation:**

Weak Accept: I recommend accepting the paper, but will not argue for my recommendation if the majority of other reviewers have a different opinion.

**Summary:**

This paper considers the online/offline construction of a dataset of challenging scenarios that could support/enhance the training of collision avoidance policies for autonomous vehicles in urban scenarios involving realistic pedestrians. In contrast to existing approaches that either construct such training scenarios out of simplified (constant-velocity motion model) or "suicidal" (collision-seeking) pedestrians, this work proposed a novel learning based method to initialize pedestrians and to train a richer set of pedestrian motion models, while still keeping the level of collision (i.e., the overall difficulty of the challenge for the AVs) high. Results are shown in a variety of realistic urban scenes, using different initialization methods and pedestrian policies. There, authors even show that, under certain assumptions, collision-avoiding pedestrians can still lead to the most collisions with AVs when initialized at the right places in a carefully arranged world.

**Issues:**

Issues were listed above in my main review (major and minor ones).

**Reviewer Expertise:**

Good: General knowledge of the area

**Strengths And Weaknesses:**

First of all, this paper is well written, and pleasant to read. I did notice a few typos or grammatical mistakes here and there (see list below), but nothing major. However, it took me multiple pages until I finally started to understand the actual premise of the paper, and I believe this is the main shortcoming of this work for now, but which can be addressed.

In my opinion, the need for non-suicidal, yet non-constant-velocity pedestrian models is currently weak in the first half of the paper. After my initial read of the abstract and introduction, I believe that I agreed with this premise (on which the entire paper relies), but would not have been able to articulate why or provide any form of example of such a scenario (i.e., non-trivial, non-suicidal pedestrian having a higher likelihood of collision with an AV).
Then, at the end of Section 3 (line 108-109), I found this sentence that finally mentions that an example of such a non-trivial scenario will be provided later in the experiments section. However, I believe this point really needs to be made clear much earlier in the text. In particular, as a reader, I believe that I would have benefited from a lot more intuition as to what class of examples of this type exist, as the paper heavily relies on this premise. For now, the paper reads a little like "Please believe us that this very interesting, non-trivial, yet counterintuitive statement is true; we will describe a method to fix it, and then at the full end provide you with one line of a table that shows that we were right (and a few sentences in the text)," and I worry this structure might not be best in supporting this premise that carries the entire paper.
Again, I can believe authors are indeed correct with their premise, which would make the work valuable and nuanced and interesting, but as a reader I would have loved to be taken in through this journey, rather than remain a spectator/believer until the "big reveal" (which also needs to be a much bigger/stronger reveal, if this is the main meat of the paper). I believe this is mostly a writing exercise, but one that will be crucial in bringing this paper to the necessary level to be understood/enjoyed by the community and be ready for publication. That is my one major concern with this paper in its current form.


Other than that, I only have a few minor comments, as the rest of the paper is very clear and articulates the method and all the assumptions made very clearly and convincingly:

1. Table 2: many of these values seem to be extremely close to each other, and I worry that these results are not statistically significant. Can authors present any form of analysis, or at least average error/standard deviations to help the reader form a more complete opinion of these results? These additional information should likely be discussed in the text as well.

2. List of typos/grammatical mistakes I caught (please perform a full spell/grammar check anyway, I am not claiming to have caught all of them of course):
	- line 154: "the \mu" --> \mu
	- line 154: 3m^{-1} --> should this rather be 3ms^{-1}? 3 1/m does not look like a velocity.
	- line 181: "encure" --> ensure?
	- line 221: "to fraction" --> to the fraction
	- line 243: could be instead be --> could instead be
	- line 253: the scenes --> the scenes'
	- line 318: mdoel --> model
	- line 321: coo-operative --> cooperative

3. The conclusion currently is very short and only summative, and therefore reads as extremely weak. It provides little over a redundant summary of the work, and no intuition or opening to future works, deeper reflection, etc. I worry it was put together rather hastily (as seen from the big typos in it), and believe it could be rewritten to better use this space to conclude the paper (key insight/take home message, deeper reflection, future works).

**Summary Of Recommendation:**

In summary, this work proposes a new method to create challenging pedestrian-vehicle scenarios for AV training using more realistic/nuanced, learned pedestrian motion models. The main premise of the paper is interesting and thought-provoking, but really needs to be expanded upon (provide intuition/example) earlier in the text, and again highlighted more thoroughly in the results section. Provided that this is addressed, I am in favor of accepting this work for presentation at CoRL 2021.

---

> ### Author Response · Authors · 2021-08-31
> **Your critique has been considered in the revision of the paper**
>
> Thank you for saying that our work is “interesting and thought-provoking”. We have thoroughly edited the paper, and hope that the revision makes a clearer point. We have added standard deviations to Table 2. As you mention a number of the pedestrian models tested in Table 2 are not significantly different from one another. However this does not change the analysis as the pedestrian models that are similar are the models based on the Semantic Pedestrian Locomotion (SPL) model. Please note that part of Table 2 was moved into the supplementary during the restructuring of the paper. The standard deviations of Table 1 are omitted as they are similar in magnitude to those of Table 2.

---

> > ### Comment · Reviewer_UAuu · 2021-09-03
> > **Response to Authors**
> >
> > Dear Authors,
> >
> > Thanks a lot for your hard work in editing and updating your manuscript! Personally speaking, you have adequately addressed my prior concerns.
> >
> > However, after reading the other reviews, I believe that some of the main concerns raised by reviewer s5dQ merit further work, at least in updating the writing of the manuscript. In particular, the "need" for a data-driven approach to this problem still seems questionable in the current text. While your work does exhibit very good results, the lack of proper comparisons with conventional approaches does not allow you to fully justify this claimed need. I would agree with reviewer s5dQ that either a few more comparison results, or at least a more nuanced discussion of the differences between data-driven and conventional approaches to this problem seems warranted (especially in light of reviewer s5dQ's detailed points on this subject). The former might need to be left as future work for now (and mentioned as such in the text), but the latter seems doable in the time between now and the camera-ready paper needs to be submitted.
> > Second, the claim that the single-pedestrian problem is actually harder than the multi-pedestrian one might need to be toned down or even removed, as reviewer s5dQ's points on the matter seem rather compelling.
> >
> > Given these two concerns are addressed, I believe this paper would be ready for publication.

---

> > > ### Author Response · Authors · 2021-09-05
> > > **Computational complexity is the limiting factor in the methods proposed by reviewer s5dQ**
> > >
> > > Dear reviewer UAuu,
> > >
> > > Thank you for reviewing our revised manuscript, and for the feedback.
> > >
> > > The main issue in comparing experimentally with the exhaustive search and the dynamic programming approach (as propsed by s5dQ) are the enormous computational costs (~233 days). Please see our reply to  the reviewer s5dQ.
> > >
> > > To clarify regarding multiple pedestrians: the one pedestrian agent case is a special case of the multi-pedestrian agent set up. Since we have shown that there exists pedestrian locations for one collision avoiding pedestrian agent that lead to collisions, then there exists also pedestrian locations for multiple pedestrians that lead to collisions even if the pedestrians are collision avoiding. The space of initial locations that lead to collisions is larger for the multi pedestrian case than in the single pedestrian case as illustrated by the scenarios a,b in s5dQ’s reply. Thus the multi-pedestrian problem has more solutions than the one-pedestrian case. Therefore the one-pedestrian case is a harder problem, but this refers only to the size of the solution space, and not how hard it is in practice to find the solution.  We are happy to formulate this to clarify that the set of solutions for the controllable multi-pedestrian case is larger than the set of solutions for one pedestrian.

---

> > > > ### Comment · Reviewer_UAuu · 2021-09-05
> > > > **Response to Authors**
> > > >
> > > > Dear Authors,
> > > >
> > > > I understand the concerns about computational cost, but as I mentioned what I was requesting were your willingness to make changes to the manuscript to support this claim that an ML approach is needed. I believe this request is reasonable given the short time frame. That is, there is no need for more experimental comparisons, although finding a way to get some would be nice for future works.
> > > >
> > > > Regarding the multi- vs single-pedestrian case, I still do not understand your point. What is your definition of a "hard" problem? In my understanding, I would say that a harder problem is one where finding the solution takes longer in average. But you seem to be comparing the size of the solution space for both problems as a way to estimate difficulty, which seems misleading/counter-intuitive, especially considering that, like you said, this is "[...] not how hard it is in practice to find the solution."
> > > > I am confused now, and would like this clarified here, as well as in the text of the paper. In general, please try to remain close to the definitions used by the community to ensure your statements are accurate and not misleading.
> > > >
> > > > In summary, for me to support this paper, I would like you to clearly state that you will make the changes in writing I have mentioned in my most recent comment, and to decsribe briefly what these changes would entail and where they would appear in the text. In particular, these changes are:
> > > > 1) a more nuanced discussion about the claimed "need" for a ML approach to the problem,
> > > > 2) a more toned down/nuanced discussion about this claim that the single-agent version of your problem is "more difficult" (to be defined and this definition motivated) than the multi-agent one.

---

> > > > > ### Comment · Reviewer_UAuu · 2021-09-06
> > > > > **Re-Reponse to Authors**
> > > > >
> > > > > PS: After further reflection, I believe I might understand your point about the single-pedestrian problem being more complex. Please confirm my thinking below is accurate, and please revisit your text to ensure it is explained clearly there as well:
> > > > >
> > > > > - The single-pedestrian problem can be seen as introducing a single constraint in the system, from which to maximize collision rate in expectation. This has to be more difficult (and therefore require a more complex/advanced approach, such as your work) than being able to introduce multiple constraints (multiple pedestrians). That is, if we take the problem in reverse, from the point of view of the AV there is only a single constraint to avoid in the single-pedestrian case, and therefore avoiding collisions would seem easier in average (and thus ensuring collisions is harder).
> > > > > - However, planning for the placement of multiple pedestrians, and their resulting dynamic which might call for join/coordinated manœuvres, is likely going to be more computationally intensive in the multi-pedestrian case.

---

> > > > > > ### Author Response · Authors · 2021-09-06
> > > > > > **Thank you for the clarifications. We are happy to adjust the paper for CR**
> > > > > >
> > > > > > Thank you for the clarifications.
> > > > > >
> > > > > > We will happily include a more detailed discussion on why an RL approach is needed in the camera ready introduction.
> > > > > >
> > > > > > Namely:
> > > > > >
> > > > > > The whole traffic scene is the complete description of the state of the environment. Traffic can contain a number of unexpected changes in the environment that can be expressed completely by an image of the scene. Specifically for a general traffic scene (when for example augmenting existing data) we cannot forecast the number of pedestrians in the scene (new pedestrians can enter the scene after the first timestep), the behaviour of each individual pedestrian, the number of cars, their location and their movement direction, and unexpected obstructions (animal or a child running on the road, an object that has fallen onto the road from a car, a collision among the external entities, change in traffic lights, etc).
> > > > > > In our problem the traffic situation is the state of the system. We cannot express the evolution of the state of the traffic deterministically and continuously. Therefore we cannot express the evolution of traffic situations as a system of differential equations. Thus we cannot use the differential game approach [49] and also we cannot use the Hamilton-Jacobi reachability [H].
> > > > > > Finally we would like to evaluate the likelihood of a collision for different initial pedestrian locations, so a likelihood would need to be additionally evaluated for each position of a reachability map.
> > > > > >
> > > > > > Bertsekas [A] suggests using the RL approach for solving Markov Decision Processes large state spaces with unknown world dynamics over classical control and planning methods to avoid the curse of dimensionality. RL approaches learn to estimate the environment dynamics needed to learn a policy of a given agent (i.e. ATS, pedestrian or AV) from the given data and/or simulations. Videos (possibly 3D) are the only complete description of the environment and thus it is preferable to use the available visual data of the scene and not approximative dynamic models for the external pedestrians and cars. It is disadvantageous to use approximative models for the external entities in planning their motion (for T timesteps), since the true motion is available from simulations and the dataset. Using approximative models in planning can be expected to lead to cumulative errors.
> > > > > >
> > > > > > Further an RL approach that generalizes across scenes does not require expensive re-calculations for previously unobserved scenes. RL methods have been used in previous work [15, 16, 17, 20, 28] for generating scenarios to stress test AVs. We are the first to reformulate the problem of AV stress testing scenario generation to allow for various pedestrian behavior models to be utilized. The scene’s semantic segmentation top view image provides insight on the common locations of real pedestrians in the scene, and on the locations that are collision prone. We are the first to utilize this insight in our model by explicitly conditioning the scenario generation model on the scene semantics. This is not a planning or a control problem. We aim to model what causes a collision when both the AV and the pedestrian agent are collision avoiding.
> > > > > >
> > > > > > Using an RL approach allows the testing of AV [C-F] and pedestrian models [31,33,38,39,40,41,42] that are conditioned on visual observations in near collision scenarios. That is the state of the environment includes the AV’s high dimensional observation of the environment and the pedestrian agent’s high dimensional observation (the SPL’s state input is 17 509 dimensional and for the HLN model 165 dimensional) of the environment.
> > > > > >
> > > > > > If a robust-control method would be used then the pedestrian would be modeled as a fictive agent including all of the noise in the environment and always taking opposing actions to the AV. This would equate to a model where the pedestrian is only trained to be adversarial to the car without any pedestrian specific behavioral constraints. This is an unrealistic model of a pedestrian, as a real pedestrian reacts to changes around itself to avoid collisions and follows traffic rules induced by the scene semantics while trying to reach some goal location. Our proposed approach allows the testing of AV models with pedestrian models that are semantically aware, collision avoiding, goal reaching and articulated. This is not possible with robust control methods.

---

> > > > > > > ### Author Response · Authors · 2021-09-06
> > > > > > > **Thank you for the clarifications. We are happy to adjust the paper for CR (continued)**
> > > > > > >
> > > > > > > And yes, you have correctly understood the claim about the single pedestrian agent vrs the multi-pedestrian agent problem from the perspective of the AV (i.e. in the multi-pedestrian problem the AV’s motion has more physical constraints, thus collision avoidance is harder for the AV and there are more possible initializations of pedestrian agents that lead to collisions than in the one-pedestrian case). This claim is not currently in the manuscript, but was only mentioned in our replies. We are happy to rephrase this claim and include it in the camera ready, removing the claim on the hardness of the problem, to instead discuss the size of the solution and the constraints imposed on the AV.
> > > > > > > The significant result being that having shown empirically that a collision inducing pedestrian initial distribution exists for the one collision avoiding pedestrian agent problem implies that such a distribution also exists in the multi pedestrian agent problem, where the pedestrian agents are collision avoiding.
> > > > > > >
> > > > > > >
> > > > > > > [A] Bertsekas, Dimitri. Reinforcement Learning and Optimal Control. Athena Scientific, 2019.
> > > > > > >
> > > > > > > [C] Abbas Sadat, Mengye Ren, Andrei Pokrovsky, Yen-ChenLin, Ersin Yumer, and Raquel Urtasun. Jointly learnable behavior and trajectory planning for self-driving vehicles. In IROS, 2019. 5, 6, 7, 8, 15
> > > > > > >
> > > > > > > [D] Wenyuan Zeng, Shenlong Wang, Renjie Liao, Yun Chen, BinYang, and Raquel Urtasun. Dsdnet: Deep structured self-driving network.CoRR, abs/2008.06041, 2020. 2, 7
> > > > > > >
> > > > > > > [E] Chen, Dian, et al. "Learning by cheating." Conference on Robot Learning. PMLR, 2020.
> > > > > > >
> > > > > > > [F] Toromanoff, Marin, Emilie Wirbel, and Fabien Moutarde. "End-to-end model-free reinforcement learning for urban driving using implicit affordances." Proceedings of the IEEE/CVF Conference on Computer Vision and Pattern Recognition. 2020.
> > > > > > >
> > > > > > > [H] Bansal, Somil, et al. "Hamilton-jacobi reachability: A brief overview and recent advances." 2017 IEEE 56th Annual Conference on Decision and Control (CDC). IEEE, 2017.

---

> > > > > > > > ### Comment · Reviewer_UAuu · 2021-09-06
> > > > > > > > **Response to Authors**
> > > > > > > >
> > > > > > > > Dear Authors,
> > > > > > > >
> > > > > > > > Thank you very much for your very complete feedback and answers to my confusion/requests. I am very satisfied with your answers.
> > > > > > > > Provided the necessary changes are also made to the final camera-ready paper, I am happy to support this work. Excellent work, and thank you for your availability and help!

---

> > > > > > > > > ### Comment · Reviewer_s5dQ · 2021-09-06
> > > > > > > > > **Very helpful discussion and good proposed paper adjustments**
> > > > > > > > >
> > > > > > > > > Dear Authors and Reviewer UAuu,
> > > > > > > > >
> > > > > > > > > Thank you so much for this discussion -- personally, I found this very helpful both in re-stating my original concerns but also in seeing the concrete changes proposed to the text. After reading the suggested additions written up by the authors, I'm pleased to say that my original concerns have been reduced. Assuming that the suggested changes will be made in the camera-ready version, I will update my suggested score to "weak accept".
> > > > > > > > >
> > > > > > > > > I'd like to thank the authors for taking a lot of their time to write up detailed responses and propose modifications to the text as well as Reviewer UAuu for steering the discussion in helpful ways. Best of luck with the the submission and I look forward to reading the camera-ready version!

---

> > > > > > > > > > ### Author Response · Authors · 2021-09-07
> > > > > > > > > > **Thank you**
> > > > > > > > > >
> > > > > > > > > > Dear reviewers UAuu and s5dQ,
> > > > > > > > > >
> > > > > > > > > >
> > > > > > > > > > Thank you for taking the time to read and reflect on our paper and our replies. We greatly appreciate your aid in improving the paper by pointing out matters that needed further clarifications, and will make the promised changes to the camera ready submission.

---

### Official Review · Reviewer_7Zap · 2021-07-27

**Originality:** Good
**Technical Quality:** Good
**Clarity Of Presentation:** Good
**Impact:** 3

**Recommendation:**

Weak Accept: I recommend accepting the paper, but will not argue for my recommendation if the majority of other reviewers have a different opinion.

**Summary:**

The paper proposes a learning-based model to generate adversarial pedestrian generation for autonomous vehicle testing. The problem is formulated as learning a pedestrian placement model such that the expected number of collisions between the pedestrian and the vehicle is maximized. REINFORCE algorithm is used to train the model by considering the pedestrian and the vehicle behavioral policies as unknown dynamics, i.e. treating the pedestrian and the vehicle as black-box models. The model learns the pedestrian initial spatial distribution based on the scene semantic, distance to the vehicle, occlusion space, and occupancy map.  The model is trained using simulated data collected from CARLA. The results show that the proposed model can result in a higher number of collisions compared to random initialization and other baselines. The sensitivity of the pedestrian placement model to the pedestrian behavior model is also studied by considering several different behavioral policies. The result shows low sensitivity in most cases.

**Issues:**

- The idea of generating challenging scenarios is interesting, but the challenging scenario seems to be narrowed down to adversarial trajectory crossing scenarios or maximizing the number of collisions. This might be a limiting assumption as it does not consider other aspects of autonomous vehicle testing. For example, a vehicle might be able to avoid a collision at the cost of sharp, sudden maneuvers. Or a vehicle might go off-road or collide with other vehicles in reaction to a pedestrian motion. In short, how to define a challenging scenario and how to evaluate an autonomous vehicle based on the test result?
- The problem formulation is convincing but the use case is still not completely clear. If the goal is to generate test cases, the result may need to be compared with manually generated test cases either heuristically or by filtering randomly generated scenarios. If the goal is to generate collision-prone smart agents then how it is better than a model predictive system that is designed to collide with another agent. Since the dynamics of a vehicle can be approximated or assumed to be known, a simple model predictive scheme might also be able to generate good initial/behavior distribution.
- The distance unit is not clear in Table 1 and 2. Is it in centimeter or meter?
- It's recommended to add a list of notations and abbreviations as there are too many abbreviations in the paper.
- Line 218, type: mdoel >> model


**Reviewer Expertise:**

Very good: Comprehensive knowledge of the area

**Strengths And Weaknesses:**

As the main strengths, the paper targets the problem of autonomous vehicle evaluation, which is an important and less studied subject in autonomous driving research. The introduction provides a convincing motivation behind the work, and the paper is generally well-written, detailed, and convincing. The supplementary materials also provide further details about the model architecture, input/output, rewards, and the algorithm implementaion. The paper seems to be an initial work that can potentially encourage further research on this subject.
The main weakness is that the approach is still limited to a single pedestrian placement as opposed to considering a variable number of pedestrians or a flow of pedestrians. The placement model is interesting, but a true test may require a combination of both pedestrian initialization and behavioral model. Since there is still a dependency on the pedestrian and vehicle behavioral policies, there is a concern that the placement model becomes dependent on those policies, contrary to being a part of an independent evaluation/test system for autonomous vehicels. Lastly, the abstract and conclusion do not seem to converge to a single problem. Abstract mentions generating maximally challenging scenarios for autonomous vehicle testing as the main problem. The paper however concludes that the pedestrian initial distribution and behavior can be learned, but how to evaluate an autonomous vehicle is still not addressed in the paper.

**Summary Of Recommendation:**

The paper is an initial work on adversarial pedestrian generation for autonomous vehicle testing. Certainly, not everything can be considered in a single paper and there is room for improvement. The main concern is that the paper is too focused on a single pedestrian initiation/behavior problem, which could also be addressed through some other techniques (e.g. model predictive scheme or some heuristic approaches) as opposed to considering a more comprehensive test generation concerning pedestrian-vehicle interactions.

---

> ### Author Response · Authors · 2021-08-31
> **On the need for learning**
>
> Please note that the problem of placing out multiple pedestrians to generate collisions is an easier problem as multiple pedestrians constrain the AV’s motion more, and there are more agents that may can collide with the AV. A single pedestrian affects its surrounding pedestrian’s motion only a little and can thus be used to augment existing data without a need to model the change in motion of the external agents.
>
> We want to find the initial pedestrian location distribution that ensures that a pedestrian with a given behaviour model collides with the AV. Note that this is not a control problem, but a learning problem. In traffic pedestrian motion varies from one individual to another. In testing the approach would be utilized with as many plausible pedestrian models as possible to stress test the AV with any possible pedestrian model. An adversarial MPC could be one of the possible pedestrian behaviour models, but as a pedestrian model it would be a poor, because pedestrian behaviour depends on a number of variables - the scene semantics, the pedestrian’s visual observation of the scene, all observable traffic entities, the pedestrian’s dynamic, the pedestrian’s personal ability to follow traffic laws. Pedestrian modelling is a well studied topic, and we propose a framework that allows us to utilize state of the art approaches from pedestrian modelling in AV testing. This ensures that the pedestrian motion that the AV is tested on is realistic, and thus gives a better estimate of what may happen in real collision scenarios than models with hand-crafted rules/dynamics. Note that the proposed model $OP\mu$ outperforms in Table 1 the prior distribution $P$ which is a heuristic based on reachability and distance to the AV.
>
> Our proposed approach allows for the suggested MPC model or any other pedestrian model that is adversarial to the AV to be utilized as a pedestrian model. Note that previous work has dealt with simultaneously optimizing the full pedestrian trajectory, but this leads only to pedestrian behaviours that are collision seeking, and thus does not reflect the true variation of possible pedestrian behaviours. We have added the Adversarial STPN model to Table 2. The Adversarial STPN model’s pedestrian behaviour is trained simultaneously with $\mu$ without any pretraining on ground truth pedestrian data and without locomotion from the Human Locomotion Network. This model closely represents the previous work. It can be seen that Adversarial STPN does not outperform the Collision avoiding SPL model. The Adversarial STPN pedestrian in difference to Collision avoiding SPL tends to zig-zag in the middle of the road to get hit by the AV.
> Most generative test case methods are dependent on the AV. AV model dependent test case generation is one of the multiple tools that should be used in true testing of AVs. Because the proposed approach is AV dependent it can in fact learn to utilize previously unknown weaknesses of the AV (i.e. a form of a black-box adversarial attack). Further the model could be trained on a basic AV model and tested on some specific AV model to ensure independence of the AV model, see Table 1 in the supplementary to see the Simultaneos-$\mu$ being tested with the base AV, eventhough the model is trained with the Simultaneos-$\rho$ AV.
>
> The problem of discomforting an AV according to a discomfort measuring cost function is easier than finding scenarios that lead to a collision between the agent and the AV , because there exist far more scenarios that discomfort the AV than there exist scenarios that cause a collision. The existing approach could easily be adopted to maximize the AV’s cost function by replacing the collision measuring term in the reward by the AV’s cost function. We chose to only consider collisions because this makes the underlying minimax problem more apparent.
>
> Finally we have added a list of mathematical notations to the supplementary material.

---

> > ### Comment · Reviewer_7Zap · 2021-09-04
> > **Response to Authors**
> >
> > Thanks for taking the time and effort in responding to my comments. I still believe that the work can be considered as initial work on a learning-based method for AV evaluation. However, I tend to disagree with the response to my main concerns in terms of single vs. multiple pedestrians, comfort vs. safety evaluation, and the evaluation aspect of the work, mainly the concern raised in the original review that "the abstract and conclusion do not seem to converge to a single problem". Reviewer s5dQ has elaborated on these points further. I am hoping the final version of the paper addresses some of these concerns before publication.

---

> > > ### Author Response · Authors · 2021-09-05
> > > **Further clarifications of the multi-pedestrian agent problem and a cost function minimizing ATS**
> > >
> > > The presented work is a first but significant step. We show that the previously unstudied problem (2) has a solution.
> > >
> > > We study the effect of the pedestrian initial location distribution on the arise of collisions in problem (2), where the pedestrian behaviour and the AV’s model are given.
> > >
> > > The purpose of our work was to find out if the pedestrian initial location has a significant effect on the AV’s collision avoidance with a pedestrian with a given behaviour model. Having shown this in the presented work we can extend the framework to multiple pedestrians, because an initial position $x_0$ that leads a single pedestrian to a collision is also a valid solution in the multi-pedestrian case when one pedestrian is placed at $x_0$ and the other pedestrians are placed at a large enough distance to the $x_0$ and the AV. Therefore if a pedestrian initial location distribution that leads to pedestrian-AV collisions exists then such a distribution must exist also for multiple pedestrians. Note that one problem being harder than another problem refers to the solution space and not to how easy or hard the solution is to find in practice. Considering multiple pedestrians requires extensive changes to the experimental framework as well as likely changes to hyperparameters, and architecture of the convolutional neural network modelling $\pi$ (see supplementary). Therefore this is a part of the planned future work.
> > >
> > > We consider collisions rather than a specific AV discomfort measuring cost function as we wished to concretize the effect of a pedestrian on the arise of collisions with an AV. Considering a different reward function (depending on the AV’s cost function) and multiple pedestrians is a separate problem. We are capable of extending the work to the problem with multiple pedestrians and a cost function, but this requires a significant amount of additional work. Since any realistic cost function will penalize collisions between the AV and the collision avoiding pedestrian agent, the set of pedestrian initial locations that lead to collisions between the AV and the pedestrian agent will be a part of the set of initial pedestrian locations that will discomfort the AV as measured by a given cost function. Our current work gives strong reason to believe that the multi-pedestrian cost-function evaluated problem is solvable with our framework.

---

> > > ### Author Response · Authors · 2021-09-05
> > > **On the abstract and the conclusion**
> > >
> > > Regarding the comments about the abstract and the conclusion; the conclusion states that the problem raised in the abstract is solvable. Please see below an outtake of the abstract and the conclusion that describe the proposed general framework (i.e. the rewriting of the AV collision stress testing as the problem of learning the collision prone initial placement distribution of the pedestrian behavior model). The outtake also summarizes the novel result that it is possible to generate collision prone scenarios even when the AV and the pedestrian model are collision avoiding.
> > >
> > > In the abstract we write (lines 8-19, the revised manuscript from uploaded in August):
> > > “In this paper we reformulate the problem as learning where to place pedestrians such that the induced scenarios are collision prone for an AV. Our pedestrian placement model can be used in conjunction with any goal driven pedestrian model -- be it hand-designed or learnable -- which makes it possible to challenge an AV with a wide range of pedestrian behaviors. Furthermore the proposed pedestrian placement model can be utilized with any goal based AV model given sufficient problem constraints... Moreover, to our best knowledge we are the first to show that it is possible to learn a collision seeking scenario generation model when both the pedestrian and the AV are collision avoiding.”
> > >
> > > In the conclusion we write (lines 302-315, the revised manuscript from uploaded in August):
> > > ”We have proposed a general framework that is capable of stress testing the collision avoidance of AVs with a wide range of pedestrian behaviors. Our empirical evaluations showed that a goal driven pedestrian model with any behavior relative to collisions can be used in this framework. This is a significant result, as it was not before obvious that a collision avoiding pedestrian model could be used to generate collisions with a collision avoiding AV. We propose the Adversarial Test Synthesizer (ATS) agent that learns the pedestrian initial locations distribution  that lead to collisions with the AV. The ATS is modelled by a neural network $\mu$ observing the top view image of the scene, the scene semantics, the occupancy of dynamic objects and the prior, and outputting a distribution of pedestrian initial locations. We showed in the experiments that $\mu$ can learn to utilize a collision avoiding pedestrian behavior model that is trained on ground truth pedestrian data and has human locomotive dynamics. This work shows that generative AV tests can utilize state of the art pedestrian motion models instead of the typically used handcrafted behavior models. Verifying AVs with state of the art pedestrian forecasting models decreases the statistical difference between the tested and real pedestrian behaviors and thus hopefully will aid in avoiding real life AV crashes. ”

---

### Author Response · Authors · 2021-09-09
**Consensus among the reviewers**

 Dear meta-reviewer,

Following the rebuttal, there is now consensus among all three reviewers. All three reviewers recommend to accept the paper after our replies. In particular we have detailed further why learning is needed in our replies to reviewers s5dQ and UAuu and both reviewers are content with the discussion. We will make the promised changes (including the further discussion on the need for learning) to the camera ready submission.

---

### Meta-Review · Area_Chair_XBoh · 2021-08-14

**Recommendation:** Accept (Poster)
**Confidence:** 5

**Metareview:**

Summary: This work proposes formulating the problem of safety validation for autonomous vehicles as generating appropriate initial positions for agents to compose difficult, potentially adversarial, scenes. It proposes using a reinforcement learning approach to learn a distribution for the initial position of a pedestrian that maximizes collisions to create challenging scenarios for an autonomous vehicle. The proposed method is validated within the CARLA simulation environment.

Clarity: The paper clearly presents an important and under-examined problem and all reviewers seemed to understand the proposed approach after reading the full paper. However, all reviewers suggest ways the paper could be reorganized to more quickly help readers grasp the core ideas.

Significance/Originality: Reviewers had mixed feedback on the impact and originality of the work. It seems like some reviewers were bought into the problem formulation whereas others wanted to see a more experimental basis.

Quality: All reviewers agreed that the paper has room for improvement in strengthening its premises (see “opportunities for improvement” below for more details).

Pros:
The paper addresses a highly critical, interesting, and under-addressed area in robotics. It was clear that all reviewers agreed this problem space is extremely important and of high interest. The paper also presents a framing of the problem that is interesting and likely to spark new discussions.

Main opportunities for improvement:
It seems like the paper has three main components - 1. a proposed framing of the safety validation problem as generating initial positions for simulated agents, 2. a decision to use RL to solve this problem, and 3. experimental results with a single pedestrian in a scene.

A major contribution of the paper is that 1 is creative and somewhat unexpected, making this paper rather unique. However, at the same time, the paper has a higher burden of convincing others to adopt that same view of the validation problem. The paper would benefit from strengthening this - specifically, consider reviewer UAuu’s feedback on how to make the “point” of the paper more crisp and upfront and reviewer s5dQ’s feedback on assumptions that need to be clarified and better supported. Reviewers 7Zap and s5dQ also both comment that the results presented involve significant simplifications (e.g. single agent, simple car model). It would be worth paying special attention to clearly delineating the scope of the problem this work attempts to solve and why that problem is “big enough”.

Regarding 2, reviewers 7Zap and s5dQ question why the proposed RL approach is needed to solve the problem given the simplifying assumptions mentioned above formulate a problem that is arguably solvable by non-ML methods. Given that using ML is a focal point for CoRL, it is necessary to clarify this point in the rebuttal.

Finally, reviewers expressed uncertainty about whether or not the technical work is large enough to be a publishable unit or is a still ultimately a “first step”. I would recommend that the rebuttal also spend time justifying why the scope of the paper represents a complete unit of research.

Thank you for considering our feedback, and we look forward to seeing the updated paper.

============= Final Decision

Thank you to the authors and reviewers for the extensive and productive discussion. After discussion, the authors have agreed to incorporate key parts of the reviewers' feedback, in particular reviewer s5dQ's concerns, in the final version of the paper, and all reviewers have achieved consensus that the paper would be a useful contribution to the conference. That being said, I think reviewer s5dQ's comments about the paper's shortcomings are valid and accepting as a poster is more appropriate.

---

> ### Author Response · Authors · 2021-08-31
> **Clarification of the generality of the proposed approach and on the validity of using a simple AV model**
>
> We provide an extensively edited revision of the paper, where in the abstract, introduction and in the conclusion we clarify the main novelties of the paper:
> - We propose a general framework to generate test scenarios for any AV model in near collision scenarios with any goal driven pedestrian motion model.
> - We model the pedestrian placement in the scenario generation on the scene semantics and the AV’s occlusion map to ensure visually and semantically feasible initial placements of pedestrians.
> - We show that modern data-driven pedestrian motion models can be utilized in generative AV test generation.
> - We study the underlying mathematical problem of a constrained minmax game that is played by the scenario generation, the AV and the pedestrian. We show the non-trivial result that if the pedestrian model is collaborating with the AV, then a solution exists for the scenario generation model.
>
> In our proposed general framework we make no assumptions about the AV beyond the assumption that the AV’s occlusion map and initial velocity are known, and that the AV’s next action can be attained given the AV’s state $s^{\rho}$. Where no assumptions are made about the state $s^{\rho}$. Similarly the pedestrian model is assumed to be goal-driven, and the pedestrian action is assumed to be attainable from the pedestrian’s state.
>
> In the experiments we show that with a basic AV model, an AV-collaborative pedestrian and with a certain set of constraints a local optima of the proposed scenario generator is learnable. We believe that this result can be extended to state of the art AV models.  From real life we can observe that collisions do occur even when a pedestrian and an AV follow traffic rules, thus we know that an initialization for the pedestrian model that induces collisions must exist. Now if the basic AV model is exchanged with a state of the art AV model, then the behaviour constraints of the problem change, and the main difficulty lies in finding the set of realistic but sufficient problem constraints such that a non-trivial solution exists. We provide insight to what such changes in the behaviour constraints may be. In particular the experiments show that structured noise modelling of the observations of the AV and the pedestrian are needed. Our paper shows that efforts in the search of appropriate behaviour constraints for a state of the art  AV is worthwhile as the proposed pedestrian initialization model is learnable. Further our work provides a model architecture and a set of problem constraints that allow for a test generator to be learnt when the pedestrian model is collision avoiding and exhibiting pedestrian behaviours learnt from real traffic scenarios.
>
> A learning approach is needed to avoid assuming that the AV’s, pedestrian’s and the external traffic entities' dynamics are known. To use a planning technique (as proposed by the reviewers)  we must further assume that the expected trajectory of the above entities is possible to evaluate. These are very constraining assumptions as it is unclear how to analytically evaluate the expected action of AV observing realistic visual inputs. Further in general if the method is used to augment existing data that may be gathered from real traffic situations then the dynamics of the external traffic agents are generally unknown. To avoid the above assumptions a learning approach must be used. We note however that if a planning approach is possible for a given AV, pedestrian model and external traffic agent  then the planning approach’s reachability map could be used as a prior that can be refined by learning. Note that the existing prior is the reachability map of a constant velocity AV and a constant velocity pedestrian, and our learnt model outperforms the prior.  Finally learning must be used to condition the pedestrian initial locations on the scene semantic and the occlusion map (both are images) without introducing possibly statistically incorrect hand-designed rules.
>
> We would like to note that considering the problem of placing out a single pedestrian to test the AV is a harder problem than placing out multiple pedestrians out of whom at least one should cause a collision for the AV. This is because multiple pedestrians constrain each other and the AV’s future motion more than a single pedestrian. Extending the existing work to multiple pedestrians is planned future work.
> Further considering only the collisions between the AV and the pedestrian is a harder problem than finding initial placements of the pedestrian that cause general discomfort for the AV.  We consider the first because this illuminates the underlying constrained minimax problem.  Any cost function that measures an AV’s discomfort will penalize collisions with a pedestrian. And thus finding pedestrian initial locations that cause the AV to be discomforted has among its solutions the set of initial pedestrian locations that lead to a collision between the AV and the pedestrian.

---

### Decision · Program_Chairs · 2021-09-13

**Decision:**

Accept (Poster)

**Comment:**

Summary: This work proposes formulating the problem of safety validation for autonomous vehicles as generating appropriate initial positions for agents to compose difficult, potentially adversarial, scenes. It proposes using a reinforcement learning approach to learn a distribution for the initial position of a pedestrian that maximizes collisions to create challenging scenarios for an autonomous vehicle. The proposed method is validated within the CARLA simulation environment.

Clarity: The paper clearly presents an important and under-examined problem and all reviewers seemed to understand the proposed approach after reading the full paper. However, all reviewers suggest ways the paper could be reorganized to more quickly help readers grasp the core ideas.

Significance/Originality: Reviewers had mixed feedback on the impact and originality of the work. It seems like some reviewers were bought into the problem formulation whereas others wanted to see a more experimental basis.

Quality: All reviewers agreed that the paper has room for improvement in strengthening its premises (see “opportunities for improvement” below for more details).

Pros:
The paper addresses a highly critical, interesting, and under-addressed area in robotics. It was clear that all reviewers agreed this problem space is extremely important and of high interest. The paper also presents a framing of the problem that is interesting and likely to spark new discussions.

Main opportunities for improvement:
It seems like the paper has three main components - 1. a proposed framing of the safety validation problem as generating initial positions for simulated agents, 2. a decision to use RL to solve this problem, and 3. experimental results with a single pedestrian in a scene.

A major contribution of the paper is that 1 is creative and somewhat unexpected, making this paper rather unique. However, at the same time, the paper has a higher burden of convincing others to adopt that same view of the validation problem. The paper would benefit from strengthening this - specifically, consider reviewer UAuu’s feedback on how to make the “point” of the paper more crisp and upfront and reviewer s5dQ’s feedback on assumptions that need to be clarified and better supported. Reviewers 7Zap and s5dQ also both comment that the results presented involve significant simplifications (e.g. single agent, simple car model). It would be worth paying special attention to clearly delineating the scope of the problem this work attempts to solve and why that problem is “big enough”.

Regarding 2, reviewers 7Zap and s5dQ question why the proposed RL approach is needed to solve the problem given the simplifying assumptions mentioned above formulate a problem that is arguably solvable by non-ML methods. Given that using ML is a focal point for CoRL, it is necessary to clarify this point in the rebuttal.

Finally, reviewers expressed uncertainty about whether or not the technical work is large enough to be a publishable unit or is a still ultimately a “first step”. I would recommend that the rebuttal also spend time justifying why the scope of the paper represents a complete unit of research.

Thank you for considering our feedback, and we look forward to seeing the updated paper.

============= Final Decision

Thank you to the authors and reviewers for the extensive and productive discussion. After discussion, the authors have agreed to incorporate key parts of the reviewers' feedback, in particular reviewer s5dQ's concerns, in the final version of the paper, and all reviewers have achieved consensus that the paper would be a useful contribution to the conference. That being said, I think reviewer s5dQ's comments about the paper's shortcomings are valid and accepting as a poster is more appropriate.